

# A non-linear Granger causality framework to investigate climate–vegetation dynamics

Christina Papagiannopoulou[1], Diego G. Miralles[2,3], Niko E. C. Verhoest[3], Wouter A. Dorigo[4], and Willem Waegeman[1]

[1]Depart. of Mathematical Modelling, Statistics and Bioinformatics, Ghent University, Belgium,
[2]Depart. of Earth Sciences, VU University Amsterdam, the Netherlands
[3]Laboratory of Hydrology and Water Management, Ghent University, Belgium
[4]Depart. of Geodesy and Geo-Information, Vienna University of Technology, Austria

*Correspondence to:* Christina Papagiannopoulou (christina.papagiannopoulou@ugent.be)

**Abstract.** Satellite Earth observation has led to the creation of global climate data records of many important environmental and climatic variables. These take the form of multivariate time series with different spatial and temporal resolutions. Data of this kind provide new means to unravel the influence of climate on vegetation dynamics. However, as advocated in this article, existing statistical methods are often too simplistic to represent complex climate-vegetation relationships due to the
assumption of linearity of these relationships. Therefore, as an extension of linear Granger causality analysis, we present a novel non-linear framework consisting of several components, such as data collection from various databases, time series decomposition techniques, feature construction methods and predictive modelling by means of random forests. Experimental results on global data sets indicate that with this framework it is possible to detect non-linear patterns that are much less visible with traditional Granger causality methods. In addition, we also discuss extensive experimental results that highlight
the importance of considering the non-linear aspect of climate–vegetation dynamics.

## 1 Introduction

Vegetation dynamics and ecosystem distribution are largely driven by the availability of light, temperature and water, thus they are mostly sensitive to climate conditions (Nemani et al., 2003; Seddon et al., 2016; Papagiannopoulou et al., *in review*). Meanwhile, vegetation also plays a crucial role in the global climate system. Plant life alters the characteristics of the atmosphere
through the transfer of water vapour, exchange of carbon dioxide, partition of surface net radiation (e.g. albedo), and impact of wind speed and direction (McPherson et al., 2007; Bonan, 2008; Nemani et al., 2003; Seddon et al., 2016; Papagiannopoulou et al., *in review*). Because of the strong two-way relationship between terrestrial vegetation and climate variability, predictions of future climate can be improved through a better understanding of the vegetation response to past climate variability.

The current wealth of Earth Observation (EO) data can be used for this purpose. Nowadays, independent sensors on different
platforms collect optical, thermal, microwave, altimetry and gravimetry information, and are used to monitor vegetation, soils, oceans and atmosphere (e.g. Su et al., 2011; Lettenmaier et al., 2015). The longest composite records of environmental and climatic variables already span up to 35 years, enabling the study of multi-decadal climate–biosphere interactions. Simple





correlation statistics using some of these data sets have led to important steps forward in understanding the links between vegetation and climate (e.g. Nemani et al., 2003; Wu et al., 2015; Barichivich et al., 2014). However, correlations in general are insufficient when it comes to assessing causality, particularly in systems like the land–atmosphere continuum in which complex feedback mechanisms are involved. A commonly used alternative consists of Granger causality modelling (Attanasio et al., 2013). Analyses of this kind have been commonly applied in climate attribution studies, to investigate the influence of one climatic variable on an another, e.g., the Granger causality of $CO_2$ on global temperature (Triacca, 2005; Kodra et al., 2011; Attanasio, 2012), vegetation and snow coverage on temperature (Kaufmann et al., 2003), sea surface temperatures on the North Atlantic Oscillation (Mosedale et al., 2006), or El Niño Southern Oscillation on the Indian monsoons (Mokhov et al., 2011). Nonetheless, Granger causality should neither be interpreted as 'real causality'; to simplify, one assumes that a time series A Granger-causes a time series B if the past of A is helpful in predicting the future of B (see Sect. 2 for a more formal definition). The underlying statistical model that is commonly considered in such a context is a linear vector autoregressive model, which is, by definition, linear – see e.g. Shahin et al. (2014); Chapman et al. (2015).

In this article we show new experimental evidence that advocates for the use of non-linear methods to study climate–vegetation dynamics, due to the non-linear nature of these interactions (Foley et al., 1998; Zeng et al., 2002; Verbesselt et al., 2016). To this end, we have assembled a vast database, comprising various global data sets of temperature, radiation and precipitation, originating from multiple online resources. We use the Normalized Difference Vegetation Index (NDVI) to characterise vegetation, which is commonly used as a proxy of plant productivity (Myneni et al., 1997). We followed an inclusive data collection approach, aiming to consider all the available data sets with a worldwide coverage, thirty-year time span and monthly temporal resolution (Sect. 3). Our novel non-linear Granger causality framework for finding climatic drivers of vegetation consists of several steps (Sect. 2). In a first step, we apply time series decomposition techniques to the vegetation and the various climatic time series to isolate seasonal cycles, trends and anomalies. Subsequently, we explore various techniques for constructing high-level features from the decomposed climatic time series. In a final step, we run a Granger causality analysis on the NDVI anomalies, while replacing traditional linear vector autoregressive models by random forests. This framework allows for modelling non-linear relationships, incorporating higher-level climatic variables and avoiding over-fitting. The results of the global application of our framework are discussed in Sect. 4.

## 2  A Granger causality framework for geosciences

### 2.1  Linear Granger causality revisited

We start with a formal introduction to Granger causality for the case of two times series, denoted as $\boldsymbol{x} = [x_1, x_2, ..., x_N]$ and $\boldsymbol{y} = [y_1, y_2, ..., y_N]$, with $N$ being the length of the time series. In this work $\boldsymbol{y}$ alludes to the NDVI time series at a given pixel, whereas $\boldsymbol{x}$ can represent the time series of any climatic variable at that pixel (e.g. temperature, precipitation or radiation). Granger causality can be interpreted as predictive causality, for which one attempts to forecast $y_t$ (at the specific time stamp $t$) given previous time stamps of $\boldsymbol{y}$ and $\boldsymbol{x}$. Granger (1969) postulated that $\boldsymbol{x}$ causes $\boldsymbol{y}$ if the forecast of $\boldsymbol{y}$ improves when information of $\boldsymbol{x}$ is taken into account. In order to make this definition more precise, it is important to introduce a performance





measure to evaluate the forecast. Below we will work with the coefficient of determination $R^2$, which is defined as follows:

$$R^2(\boldsymbol{y}, \hat{\boldsymbol{y}}) = 1 - \frac{RSS}{TSS} = 1 - \frac{\sum_{i=P+1}^{N}(y_i - \hat{y}_i)^2}{\sum_{i=P+1}^{N}(y_i - \bar{y})^2} \tag{1}$$

where $\boldsymbol{y}$ represents the observed time series, $\bar{y}$ is the mean of this time series, $\hat{\boldsymbol{y}}$ is the predicted time series obtained from a given forecasting model, and $P$ is the length of the lag-time moving window. Therefore, the $R^2$ can be interpreted as the

fraction of explained variance by the forecasting model, and it increases when the performance of the model increases, reaching the theoretical optimum of 1 for an error-free forecast, and being negative when the predictions are less representative of the observations than the mean of the observations. Using $R^2$, one can now define Granger causality in a more formal way.

**Definition 1.** *We say that time series $\boldsymbol{x}$ Granger-causes $\boldsymbol{y}$ if $R^2(\boldsymbol{y}, \hat{\boldsymbol{y}})$ increases when $x_{t-1}, x_{t-2}, ..., x_{t-P}$ are considered for predicting $y_t$, in contrast to considering $y_{t-1}, y_{t-2}, ..., y_{t-P}$ only, where $P$ is the lag-time moving window.*

In climate sciences, linear vector autoregressive (VAR) models are often employed to make forecasts (Stock and Watson, 2001; Triacca, 2005; Kodra et al., 2011; Attanasio, 2012). A linear VAR model of order $P$ boils down to the following representation:

$$\begin{bmatrix} y_t \\ x_t \end{bmatrix} = \begin{bmatrix} \beta_{01} \\ \beta_{02} \end{bmatrix} + \sum_{p=1}^{P} \begin{bmatrix} \beta_{11p} & \beta_{12p} \\ \beta_{21p} & \beta_{22p} \end{bmatrix} \begin{bmatrix} y_{t-p} \\ x_{t-p} \end{bmatrix} + \begin{bmatrix} \epsilon_1 \\ \epsilon_2 \end{bmatrix}$$

with $\beta_{ij}$ being parameters that need to be estimated and $\epsilon_1$ and $\epsilon_2$ referring to two white noise error terms. This model can be

used to derive the predictions required to determine Granger causality. In that sense, time series $\boldsymbol{x}$ Granger-causes time series $\boldsymbol{y}$ if at least one of the parameters $\beta_{12p}$ for any $p$ significantly differs from zero. Specifically, and since we are focusing only on the vegetation time series as target time series, the following two models are compared:

$$y_t \approx \hat{y}_t = \beta_{01} + \sum_{p=1}^{P} \left( \beta_{11p} y_{t-p} + \beta_{12p} x_{t-p} \right) \tag{2}$$

$$y_t \approx \hat{y}_t = \beta_{01} + \sum_{p=1}^{P} \beta_{11p} y_{t-p} \tag{3}$$

We will refer to model (2) as the *full model* and to model (3) as the *baseline model*, since the former incorporates all available information and the latter only information of $\boldsymbol{y}$.

Comparing the above two models, $\boldsymbol{x}$ Granger-causes $\boldsymbol{y}$ if the full model manifests a substantially-better predictive performance than the baseline model. To this end, statistical tests can be employed, for which one typically assumes that the errors in the model follow a Gaussian distribution (Maddala and Lahiri, 1992). However, our above definition differs from the perspec-

tive in papers that develop statistical tests for Granger causality (Hacker and Hatemi-J, 2006), because we intend to move away from statistical hypothesis testing, since the assumptions behind such testing are typically violated when working with climate data where neither variables nor observational techniques are fully independent in most cases, and errors are not normally distributed.

In climate studies, the causal relationship between two time series $\boldsymbol{x}$ and $\boldsymbol{y}$ has often been investigated in the bivariate setting

(Elsner, 2006, 2007; Kodra et al., 2011; Attanasio, 2012; Attanasio et al., 2012). However, such an analysis might lead to wrong





conclusions, because additional (confounding) effects exerted by other climatic or environmental variables are not taken into account (Geiger et al., 2015). This problem can be solved by considering time series of additional variables. For example, let us assume one has observed a third variable $w$, which might act as a confounder in deciding whether $x$ Granger-causes $y$. The above definition then naturally extends as follows.

**Definition 2.** *We say that time series $x$ Granger-causes $y$ conditioned on time series $w$ if $R^2(y, \hat{y})$ increases when $x_{t-1}, x_{t-2}, ...,$ $x_{t-P}$ are considered for predicting $y_t$, in contrast to considering $y_{t-1}, y_{t-2}, ..., y_{t-P}$ and $w_{t-1}, w_{t-2}, ..., w_{t-P}$ only, where $P$ is the lag-time moving window.*

An extension of this definition for more than three times series is straightforward. In the tri-variate setting, Granger causality might be tested using the following linear VAR model:

$$
\begin{bmatrix} y_t \\ x_t \\ w_t \end{bmatrix} = \begin{bmatrix} \beta_{01} \\ \beta_{02} \\ \beta_{03} \end{bmatrix} + \sum_{p=1}^{P} \begin{bmatrix} \beta_{11p} & \beta_{12p} & \beta_{13p} \\ \beta_{21p} & \beta_{22p} & \beta_{23p} \\ \beta_{31p} & \beta_{32p} & \beta_{33p} \end{bmatrix} \begin{bmatrix} y_{t-p} \\ x_{t-p} \\ w_{t-p} \end{bmatrix} + \begin{bmatrix} \epsilon_1 \\ \epsilon_2 \\ \epsilon_3 \end{bmatrix}, \tag{4}
$$

where a causal relationship between $x$ and $y$ exists if at least one $\beta_{12p}$ significantly differs from zero.

## 2.2 Over-fitting and out-of-sample testing

It is well known in the statistical literature that predictions made on in-sample data, that is, data used to fit a statistical model, tend to be optimistic. This process is often referred to as over-fitting, i.e., by definition, the fitting process chooses parameters that mimic the data as closely as possible (Friedman et al., 2001). In the context of Granger causality analysis, over-fitting will occur more prominently in the multivariate case, when the number of considered time series increases. In the experimental analysis that will be presented in Sect. 4, we will consider in total 21 time series, resulting in $21(P+1)$ different parameters for each equation of the VAR model. Simple models of that kind will as a result already be vulnerable to over-fitting. The situation further aggravates when switching from linear to non-linear models, because then the number of parameters typically increases since the general form of the model is unknown.

To prevent over-fitting, out-of-sample data should be used in evaluating the predictive performance in Granger causality studies (Gelper and Croux, 2007). The most straightforward procedure for creating out-of-sample data is to separate the time frame into two parts, a training set and a test set, which typically constitute the first and last half of the time frame. A few authors have adopted this approach for climate attribution (Attanasio et al., 2012; Pasini et al., 2012); however, satellite EO time series are usually too short to allow for train-test splitting in that fashion. An alternative approach, which uses the available data in an efficient manner, is cross-validation. To this end, the time frame is divided in a number of short intervals, typically a few years of data, in which one interval serves as a test set, while all remaining data are used for parameter fitting. This procedure is repeated until all intervals have served once as a test set, and the prediction errors obtained in each round are aggregated, so that one global performance measure can be computed. We direct the reader to Von Storch and Zwiers (2001) and Michaelsen (1987) for further discussion.





The inclusion of a regularization term in the fitting process of over-parameterized linear models will avoid over-fitting. Typical regularizers that shrink the parameter vectors of linear models towards zero are L2-norms as in ridge regression, L1-norms as in Least Absolute Shrinkage and Selection Operator (LASSO) models, or a combination of the two norms, as in elastic nets (Friedman et al., 2001). Translated to VAR models, this implies that one should impose restrictions on the parameter matrix of Eq. (4), as done in recent theoretical papers (Gregorova et al., 2015). In this work, we want to identify causal relationships between a vegetation time series and various climatic time series, for which a simpler approach can be adopted, since the target variable, in our case is only one. Denoting the vegetation time series by $y$, one can mimic in the tri-variate setting a VAR model by means of three autoregressive ridge regression models:

$$y_t \quad \approx \quad \hat{y}_t = \beta_{01} + \sum_{p=1}^{P} \left( \beta_{11p}y_{t-p} + \beta_{12p}x_{t-p} + \beta_{13p}w_{t-p} \right) \tag{5}$$

$$x_t \quad \approx \quad \hat{x}_t = \beta_{02} + \sum_{p=1}^{P} \left( \beta_{21p}y_{t-p} + \beta_{22p}x_{t-p} + \beta_{23p}w_{t-p} \right) \tag{6}$$

$$w_t \quad \approx \quad \hat{w}_t = \beta_{03} + \sum_{p=1}^{P} \left( \beta_{31p}y_{t-p} + \beta_{32p}x_{t-p} + \beta_{33p}w_{t-p} \right) \tag{7}$$

In this article we aim to detect the climate drivers of vegetation, and not the feedback of vegetation on climate (Green and et al., *in review*). Therefore, it suffices to retain Eq. (5) in our analysis as is stated above for the bivariate case (Eq. 3). Concatenating all parameters of this model into a vector $\boldsymbol{\beta} = [\beta_{01}, \beta_{111}, ..., \beta_{13p}]$, one fits in ridge regression the parameters by solving the following optimization problem:

$$\min_{\boldsymbol{\beta}} \sum_{P+1}^{N} (y_t - \hat{y}_t)^2 + \lambda ||\boldsymbol{\beta}||^2,$$

with $\lambda$ being a regularization parameter, that is tuned using a validation set or nested cross-validation and $||\boldsymbol{\beta}||^2$ being a penalty term, i.e. the squared $\ell_2$ norm of the coefficient vector. The sum only starts at $P+1$ because a moving window of $P$ lags is considered. For simplicity we describe the above approach for the tri-variate setting, even though the total number of variables used in our study amounts to 3,884; nonetheless, extensions of this the multivariate setting are straightforward.

## 2.3 Non-linear Granger causality

The methodology that we develop in this paper is closely connected to the methods explained in the previous section. However, as we hypothesize that the relationships between climate and vegetation can be highly non-linear (Foley et al., 1998; Zeng et al., 2002; Verbesselt et al., 2016), we also replace the linear VAR-models in the Granger causality framework with non-linear machine learning models. The machine learning algorithm we choose is random forests (Breiman, 2001), a well-known method that has shown its merits in diverse application domains, and that has successfully been applied to EO observations in both classification and regression problems (Dorigo et al., 2012; Rodriguez-Galiano et al., 2012; Loosvelt et al., 2012a, b). Briefly summarized, the random forest algorithm forms a combination of multiple decision trees, where each tree contributes with a single vote to the final output, which is the most frequent class (for classification problems) or the average (for regression problems).



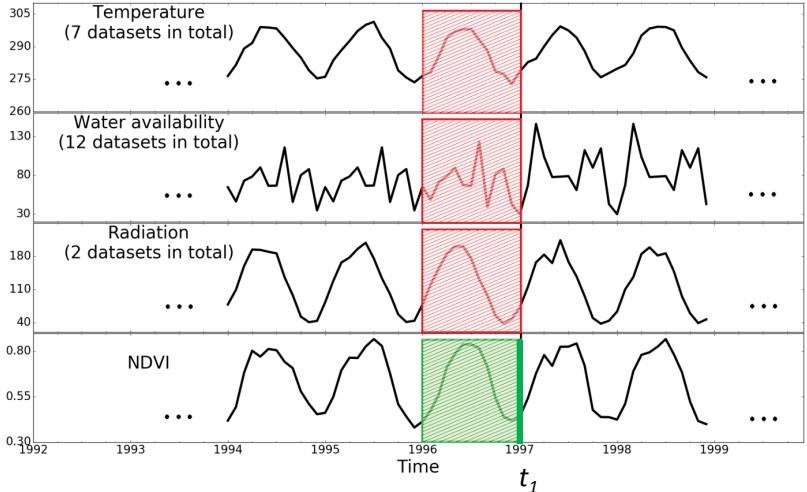

**Figure 1.** An illustrative example of the moving window approach considered in the analysis of vegetation drivers at a given time stamp $t_1$. NDVI takes here the role of the time series $\boldsymbol{y}$ in Eq. 2. In addition three climate predictor time series are shown. The baseline random forest model only considers the green moving window, whereas the full random forest model includes the red moving windows as well. The pixel corresponds to a location in North America (lat: 37.5, long: -87.5).

However, compared to most application domains where random forests are applied, we employ the algorithm in a slightly different way: as an autoregressive method for time series forecasting, similar to VAR models but non-linear. In practice this means that we simply replace the full and baseline linear model of Sect. 2.1 by a random forest model. At each pixel the vegetation time series will still be considered as response variable, and various climate time series serve as predictor

5     variables (see Sect. 3.1 for an overview of our database). For a given value of the NDVI time series $\boldsymbol{y}$ at time stamp $t$, we investigate properties of the different predictor time series – i.e., temperature, radiation, etc. – by considering a moving window including a number of previous months (Fig.1). In this way, the definition of Granger causality in Sect. 2.1 is adopted. Any climatic time series $\boldsymbol{x}$ Granger-causes vegetation time series $\boldsymbol{y}$ if the predictive performance in terms of $R^2$ improves when the moving window $x_{t-1}, x_{t-2}, ..., x_{t-P}$ is incorporated in the random forest, in contrast to considering $y_{t-1}, y_{t-2}, ..., y_{t-P}$

10     and $w_{t-1}, w_{t-2}, ..., w_{t-P}$ only. Analogous to the linear case, we will refer to the full random forest model when all variables are taken into account and to the baseline random forest model when only the moving window $y_{t-1}, y_{t-2}, ..., y_{t-P}$ of $\boldsymbol{y}$ is considered. In Fig. 1 this principle is extended to four time series. The baseline random forest predictions of NDVI at $t_1$ are based on the observations from the green moving window only, whereas the full random forest model includes the three red moving windows as well when predicting the NDVI at $t_1$.

15     In our experiments we treat each continental pixel as a separate problem, and use the implementation of scikit-learn (Pedregosa et al., 2011) for the random forest regressor, setting the number of trees equal to 100 and the maximum number of predictor variables per node to the square root of the total number of predictor variables. Changes in these parameters (or the





randomness of the algorithm) did not cause substantial changes in the results (not shown). The model was assessed by means of five-fold cross-validation. The window length was fixed to twelve months because initial experimental results revealed that longer time windows did not lead to improvements in the predictions (results omitted). We also experimented with techniques that exploit spatial correlations to improve the predicted performance of the model (see Sect. 5.1).

## 3 Database creation and variable construction

### 3.1 Global data sets

Our non-linear Granger causality framework is used to disentangle the effect of past-time climate variability on global vegetation dynamics. To this end, climate data sets of observational nature – mostly based on satellite and *in situ* observations – have been assembled to construct time series (see Sect. 3.3) that are then used to predict NDVI anomalies following the linear and non-linear causality frameworks described in Sect. 2. Data sets have been selected from the current pool of satellite and *in situ* observations on the basis of meeting a series of spatio-temporal requirements: (a) expected relevance of the variable for driving vegetation dynamics, (b) multi-decadal record and global coverage available, and (c) adequate spatial and temporal resolution. The selected data sets can be classified into three different categories: water availability (including precipitation, snow water equivalent and soil moisture data sets), temperature (both for the land surface and the near-surface atmosphere), and radiation (considering different radiative fluxes independently). Rather than using a single data set for each variable, we have collected all data sets meeting the above requirements. This has led to a total of twenty-one different data sets which are listed in Table 1. They span the study period 1981–2010 at the global scale, and have been converted to a common monthly temporal resolution and $1° \times 1°$ latitude-longitude spatial resolution. To do so, we have used averages to re-sample original data sets found at finer native resolution, and linear interpolation to resample coarser-resolution ones.

For temperature we consider seven different products based on *in situ* and satellite data: Climate Research Unit (CRU-HR) Harris et al. (2014), University of Delaware (UDel) (Willmott et al., 2001), NASA Goddard Institute for Space Studies (GISS) (Hansen et al., 2010), Merged Land-Ocean Surface Temperature (MLOST) (Smith et al., 2008), International Satellite Cloud Climatology Project (ISCCP) (Rossow and Duenas, 2004), and Global Land Surface Temperature Data (LST) (Coccia et al., 2015). We also included one reanalysis data set, the European Centre for Medium-Range Weather Forecasts (ECMWF) ERA-Interim (Dee et al., 2011). In the case of precipitation, eight products have been collected. Four of them result from the merging of *in-situ* data only: Climate Research Unit (CRU-HR) (Harris et al., 2014), University of Delaware (UDel) (Willmott et al., 2001), Climate Prediction Center Unified analysis (CPC-U) (Xie et al., 2007), and the Global Precipitation Climatology Centre (GPCC) (Schneider et al., 2008). The rest result from a combination of *in-situ* and satellite data, and may include reanalysis: CPC Merged Analysis of Precipitation (CMAP) (Xie and Arkin, 1997), ERA-Interim (Dee et al., 2011), Global Precipitation Climatology Project (GPCP) (Adler et al., 2003), and Multi-Source Weighted-Ensemble Precipitation (MSWEP) (Beck et al., 2016). For radiation two different products have been collected; first the NASA Global Energy and Water cycle Exchanges (GEWEX) Surface Radiation Budget (SRB) (Stackhouse et al., 2004) based on satellite data, and the second one the ERA-Interim reanalysis (Dee et al., 2011). For soil moisture we use the Global Land Evaporation Amsterdam Model (GLEAM)





**Table 1.** Data sets used in our experiments. Basic data set characteristics are provided, including the native spatial and temporal resolutions and the primary data source.

| Variable | Product Name | Spatial Resolution | Temporal Resolution | Primary data source | Reference |
|---|---|---|---|---|---|
| Temperature | CRU-HR | 0.5° | monthly | *in situ* | Harris et al. (2014) |
| | UDel | 0.5° | monthly | *in situ* | Willmott et al. (2001) |
| | ISCCP | 1° | daily | satellite | Rossow and Duenas (2004) |
| | ERA-Interim | 0.25° | daily | reanalysis | Dee et al. (2011) |
| | GISS | 2° | monthly | *in situ* | Hansen et al. (2010) |
| | MLOST | 5° | monthly | *in situ* | Smith et al. (2008) |
| | LST | 0.5° | daily | satellite | Coccia et al. (2015) |
| Water availability | CRU-HR | 0.5° | monthly | *in situ* | Harris et al. (2014) |
| | MSWEP | 0.25° | daily | satellite/*in situ* | Beck et al. (2016) |
| | UDel | 0.5° | monthly | *in situ* | Willmott et al. (2001) |
| | CMAP | 2.5° | monthly | satellite/*in situ* | Xie and Arkin (1997) |
| | CPC-U | 0.25° | daily | *in situ* | Xie et al. (2007) |
| | GPCC | 0.5° | monthly | *in situ* | Schneider et al. (2008) |
| | GPCP | 2.5° | monthly | satellite/*in situ* | Adler et al. (2003) |
| | ERA-Interim | 1° | daily | reanalysis | Dee et al. (2011) |
| | GLEAM | 0.25° | daily | satellite | Miralles et al. (2011) |
| | ESA CCI-PASSIVE | 0.25° | daily | satellite | Liu et al. (2012) |
| | ESA CCI-COMBINED | 0.25° | daily | satellite | Liu et al. (2012) |
| | GLOBSNOW | 0.25° | daily | satellite | Luojus et al. (2010) |
| Radiation | SRB | 1° | daily | satellite | Stackhouse et al. (2004) |
| | ERA-Interim | 0.25° | daily | reanalysis | Dee et al. (2011) |
| Greenness (NDVI) | GIMMS | 0.25° | monthly | satellite | Tucker et al. (2005) |

(Miralles et al., 2011), and the Climate Change Initiative (CCI) product (Liu et al., 2012, 2011); two different soil moisture products by CCI are considered: the passive microwave dataset and the combined active/passive product (Dorigo et al., *in review*). Moreover, snow water equivalents data come from the GlobSnow project (Luojus et al., 2010).

To conclude, as a proxy for the state and activity of vegetation, we use the third generation (3G) Global Inventory Modeling and Mapping Studies (GIMMS) satellite-based NDVI (Tucker et al., 2005), a commonly used long-term global record of NDVI (Beck et al., 2011). We note that this dataset is used to derive the response variable in our approach (seasonal NDVI anomalies, see Sect. 3.2), while all other data sets are converted to predictor variables. The length of the NDVI record (1981–2010) sets the study period to an interval of 30 years.



## 3.2 Anomaly decomposition

In climate studies, it is common to apply Granger causality analyses on time series of seasonal anomalies. The latter may be obtained in a two-step decomposition procedure, by first subtracting the seasonal cycle and then the long-term trend from the raw time series (Attanasio, 2012). Several competing decomposition methods have been proposed in the literature, including additive models, multiplicative models and more sophisticated methods based on break points (see e.g. Cleveland et al., 1990; Verbesselt et al., 2010; Grieser et al., 2002). In our framework we used the following approach: In a first step, at each given pixel, the 'raw' time series of the target variable $y_t$ and the climate predictors ($x_t$, $w_t$,...) are de-trended linearly based on a simple linear regression with the time stamp $t$ as predictor variable applied to the entire study period. For the case of the target variable this can be denoted as follows:

$$y_t \approx y_t^T = \alpha_0 + \alpha_1 t,$$

with $\alpha_0$ and $\alpha_1$ being the intersect and the slope of the linear regression, respectively. We obtain in this way the de-trended time series $y_t^D = y_t - y_t^T$. This de-trending is needed to remove obvious non-stationary signals in climatic time series, and allows us to draw the emphasis to the shorter-term multi-month dynamics. By de-trending one can assure that the mean of the probability distribution does not change over time; however, other moments of the probability distribution, such as the variance, might still be time-dependent. As classical statistical procedures for testing Granger causality (i.e. autoregressive model, statistical tests) are developed for stationary time series, those methods are in fact not applicable to non-stationary climate data. In a second step, after subtracting the trend from the raw time series, the seasonal cycle $y_t^S$ is calculated. When the assumption is made that the seasonal cycle is annual and constant over time, one can simply estimate it as the monthly expectation. To this end, the multi-year average for each of the twelve months of the year ($y_t^S$) is calculated. Finally, the anomalies $y_t^R$ can then be computed by subtracting the corresponding monthly expectation from the de-trended time series: $y_t^R = y_t^D - y_t^S$. This procedure is schematically represented in Fig. 2.

## 3.3 Predictor variable construction

We do not limit our approach to considering raw versus anomaly time series of the data sets in Table 1 as predictors, but also take into consideration different lag times, past-time cumulative values and extreme indices calculated based on these raw and anomaly time series. Our application of Granger causality can be interpreted as a way to identify patterns in climate during past-time moving windows (see Fig. 1) that are predictive with respect to the anomalies of vegetation time series. Therefore, by feeding predictor variables from previous time stamps to a linear (or non-linear) predictive model, one can identify sub-sequences of interest in the moving window specified for time stamp $t$, a technique that is similar to so-called shapelets (Ye and Keogh, 2009). Techniques for finding shapelets have been mainly applied to the problem of time series classification, where they are used to extract meaningful information from raw time series (Grabocka et al., 2014; Rakthanmanon and Keogh, 2013; Zakaria et al., 2012; Ye and Keogh, 2009; Mueen et al., 2011; Lines et al., 2012; Hills et al., 2013). In our case, vegetation dynamics may not necessarily reflect the climatic conditions from (e.g.) three months ago, but the average of the (e.g.) three antecedent months. This integrated response to antecedent environmental and climatic conditions is referred here as



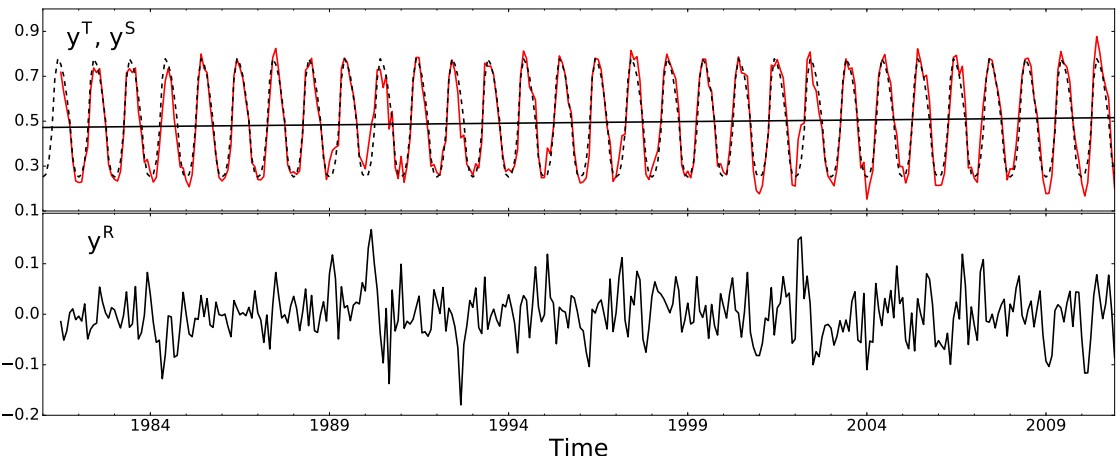

**Figure 2.** The three components of the NDVI time series decomposition of a specific pixel of the northern hemisphere (lat: 53.5, long: 26.5). On top, the linear trend and the seasonal cycle fitted on the raw data, on the bottom the remaining anomalies. All three components of the time series are shown in black, whereas the raw time series is shown in red. See text for details.

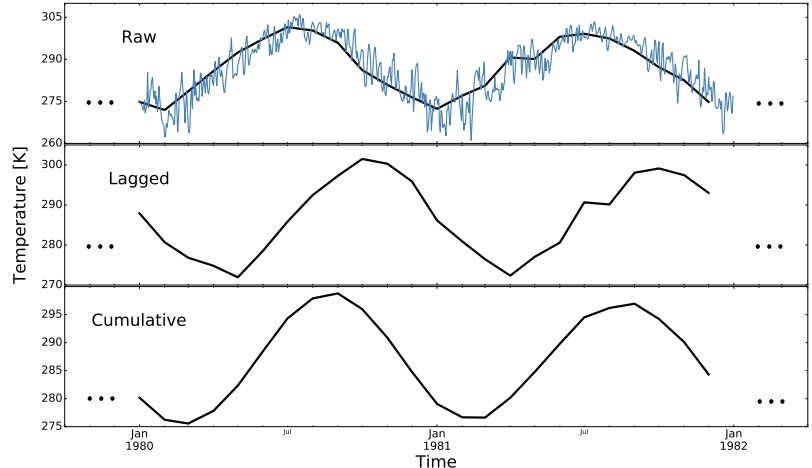

**Figure 3.** Example of lagged and cumulative variables extracted from a temperature time series. On top, part of a raw daily time series with its monthly aggregation. In the middle, the 4-month lag-time monthly time series is presented, while on the bottom, the corresponding 4-month cumulative variable. The pixel corresponds to a location in North America (lat: 37.5, long: -87.5)





**Table 2.** Extreme indices considered as predictive variables. These indices are derived from the raw (daily) data and the (daily) anomalies of the data sets in Table 1. We also calculate the lagged and cumulative variables from these extreme indices.

| Name | Description |
| --- | --- |
| STD | Standard deviation of daily values per month |
| DIR | Difference between max and min daily value per month |
| Xx | Max daily value per month |
| Xn | Min daily value per month |
| Max5day | Max over 5 consecutive days per month |
| Min5day | Min over 5 consecutive days per month |
| X99p/X95p/X90p | Number of days per month over $99^{th}$/$95^{th}$/$90^{th}$ percentile |
| X1p/X5p/X10p | Number of days per month under $1^{th}$/$5^{th}$/$10^{th}$ percentile |
| T25C[a] | Number of days per month over $25°$C |
| T0C[a] | Number of days per month below $0°$C |
| R10mm/R20mm[b] | Number of days per month over 10/20 mm |
| CHD (Consecutive High value Days) | Number of consecutive days per month over $90^{th}$ percentile |
| CLD (Consecutive Low value Days) | Number of consecutive days per month below $10^{th}$ percentile |
| CDD (Consecutive Dry Days) [b] | Number of consecutive days per month when precipitation < 1 mm |
| CWD (Consecutive Wet Days)[b] | Number of consecutive days per month when precipitation $\geq$ 1 mm |
| Spatial Heterogeneity [c] | Difference between max and min values within 1 degree box |

[a]Only for temperature data sets

[b]Only for precipitation data sets

[c]Only for data sets with native spatial resolution <1° lat-lon

a 'cumulative' response. More formally, we construct a cumulative variable of $k$ months as the sum of time series observations in the last $k$ months:

$$\text{Cum}[x_{t-1}, x_{t-2}, ..., x_{t-k}] = \sum_{p=1}^{k} x_{t-p}.$$

Note that, unlike in the case of lagged variables, cumulative ones include always the period up to time $t$. Figure 3 illustrates an example of a 4-month cumulative variable. In our analysis we have experimented with time lags covering a wide range of time-lag values, concluding that including lags of more than six months did not yield substantial predictive power.

Another type of higher-level predictor variables that can be constructed from the data sets in Table 1 are extreme indices. Over the last few years, many research studies have focused on identifying climate extremes (Nicholls and Alexander, 2007; Zwiers et al., 2013), even though fewer have concentrated on the effect of climate extremes on vegetation (Reyer et al., 2013; Smith, 2011; Zscheischler et al., 2014a, b, c). The Expert Team on Climate Change Detection and Indices (ETCCDI) recommends the use of a range of extreme indices related to temperature and precipitation (Donat et al., 2013; Zhang et al., 2011). Here we





calculate a variety of analogous indices, based on both the raw data sets as well as on the seasonal anomalies (see Table 2). In addition, we derived lagged and cumulative predictor variables from these extremes indices as described above; this is done to incorporate the potential impact of climatic extremes occurring (e.g.) three months ago, or during the previous (e.g.) three months, respectively. All these resulting time series appear as additional predictor variables in our non-linear Granger causality framework (see Sect. 2.3).

Combining the different climate and environmental predictor variables described above, we obtain a database of 3,884 predictor variables per 1° pixel, covering thirty years at a monthly temporal resolution.

## 4 Results

### 4.1 Detecting linear Granger-causal relationships

In a first experiment, we evaluate the extent to which climate variability Granger-causes the anomalies in vegetation using a standard Granger causality approach, in which only linear relationships between climate (predictors) and vegetation (target variable) are considered. To this end, ridge regression is used as a linear vector autoregressive (VAR) model in the Granger causality approach (note this ridge regression will be substituted by our non-linear random forest approach in Sect. 4.2). In the application of the ridge regression, we use all climatic and environmental predictor variables (Sect. 3.3), and adopt a nested 5-fold cross-validation to properly tune the hyper-parameter $\lambda$. Figure 4a shows the predictive performance of the full ridge regression model. While the model explains more than 40% of the variability in NDVI anomalies in some regions ($R^2 >$ 0.4), this is by itself not necessarily indicative of climate Granger-causing the vegetation anomalies, as it may reflect simple correlations. In order to test the latter, we compare the results of the full model to a baseline model, i.e., an autoregressive ridge regression model that only uses previous values of NDVI to predict the NDVI at time $t$ (see Sect. 2.1). If climate Granger-caused the variability of NDVI at a given pixel, the full ridge regression model (Fig. 4a) would show an increase in the predictive power over the predictions based on the baseline ridge regression model. However, results unequivocally show that – when only linear relationships between vegetation and climate are considered – the areas for which vegetation anomalies are Granger-caused by climate are very limited, including mainly semiarid regions and central Europe (Fig. 4b).

For further comparison, we analyze the predictive performance obtained when (linear) Pearson correlation coefficients are calculated on the training data sets, selecting the highest correlation to the target variable for any of the 3,884 predictor variables at each pixel. Figure 4c shows that the explained variance is again rather low, and for most regions substantially lower than the $R^2$ of the baseline ridge regression model, here considered as the minimum to interpret this predictive power as Granger-causal. These results indicate that, despite being routinely used as a standard tool in many climate–biosphere studies (see e.g. Nemani et al., 2003; Wu et al., 2015), univariate correlation analyses are unable to extract the nuances of the relationships between climate and vegetation dynamics.





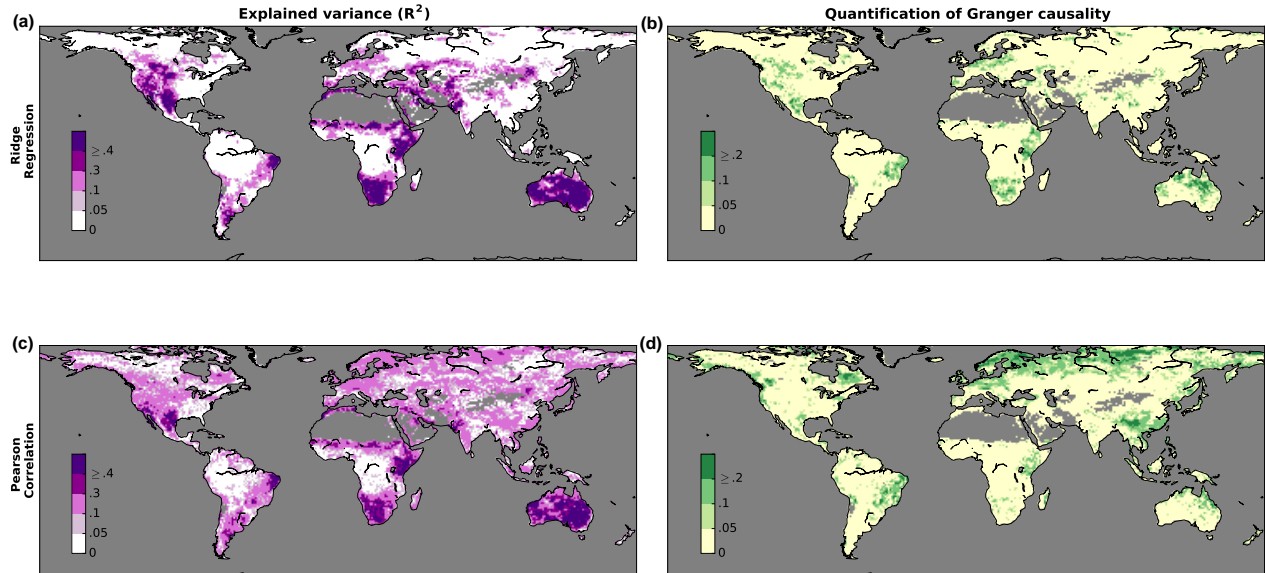

**Figure 4.** Linear Granger causality of climate on vegetation. **(a)** Explained variance ($R^2$) of NDVI anomalies based on a full ridge regression model in which all climatic variables are included as predictors. **(b)** Improvement in terms of $R^2$ by the full ridge regression model with respect to a baseline ridge regression model that uses only past values of NDVI anomalies as predictors; positive values indicate (linear) Granger causality. **(c)** A filter approach in which the variable with the highest squared Pearson correlation against the NDVI anomalies is selected. **(d)** Improvement in terms of $R^2$ by the filter approach with respect to a the same baseline ridge regression model that uses only past values of NDVI anomalies only.

## 4.2 Linear versus non-linear Granger causality

To analyze the effect of climate on vegetation more thoroughly, we substitute the linear ridge regression model (VAR) by our non-linear random forest model. Results in Fig. 5 highlight the differences. Compared to the results in the previous Sect. 4.1, the predictive power substantially increases by considering non-linear relationships between vegetation and climate (Fig. 5a). This is the case for most land regions, but is especially remarkable in semiarid regions of Australia, Africa, Central and North America, which are frequently exposed to water limitations. In those regions, more that 40% the variance of NDVI anomalies can be explained by antecedent climate variability. These results are further investigated by Papagiannopoulou et al. (*in review*), who highlight the crucial role of water supply for the anomalies in vegetation greenness in these regions. On the other hand, the variance of NDVI explained in other regions such as the Eurasian taiga, tropical rainforests or China is again below 10%. We hypothesize two potential reasons: (a) the uncertainty in the observations used as predictor are typically larger in these regions (especially in tropical forests and at higher latitudes) (Dorigo et al., 2010), and (b) these are regions in which



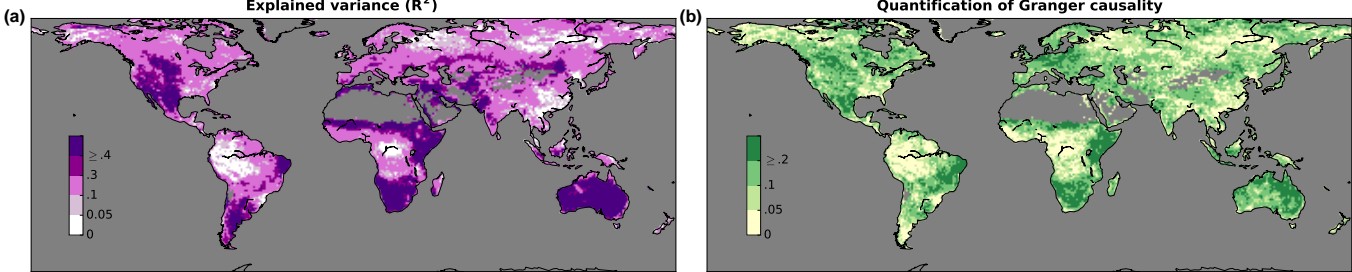

**Figure 5.** Non-linear Granger causality of climate on vegetation. **(a)** Explained variance ($R^2$) of NDVI anomalies based on a full random forest model in which all climatic variables are included as predictors. **(b)** Improvement in terms of $R^2$ by the full random forest model with respect to the baseline random forest model that uses only past values of NDVI anomalies as predictors; positive values indicate (non-linear) Granger causality.

vegetation is not necessarily primarily controlled by climate, but may also be driven by phenological and biotic factors (Hutyra et al., 2007), occurrence of wild fires (Van der Werf et al., 2010), limitations imposed by the availability of soil nutrients (Fisher et al., 2012) or agricultural practices (Liu et al., 2015). Nonetheless, the patterns of explained variance shown in Fig. 5a are again not necessarily indicative of Granger causality. As we did in Fig. 4b, in order to test whether the climatic and environmental controls do, in fact, Granger-cause the vegetation anomalies, we compare the results of our full random forest model to a baseline random forest model which only uses previous values of NDVI to predict the NDVI at time $t$ (see Fig. 5b). In this case, the improvement over the baseline is unambiguous. One can conclude that – while not bearing into consideration all potential control variables in our analysis – climate dynamics indeed Granger-cause vegetation anomalies in most of the continental land surface, being larger their impact on subtropical regions and mid-latitudes. Moreover, a comparison between Figs. 4b and 5b unveils that these causal relationships are highly non-linear as expected, given the progressive response of vegetation to environmental changes and the recovery time of vegetation to these perturbations (Foley et al., 1998; Zeng et al., 2002; Verbesselt et al., 2016).

# 5 Discussion

## 5.1 Spatial and temporal aspects

Environmental dynamics reveal their effect on vegetation at different time scales. Since the adaptation of vegetation to environmental changes requires some time, and because soil and atmosphere have a memory, a necessary aspect to investigate is the potential lag-time response of vegetation to climate dynamics. The idea of exploring lag times has been introduced by several studies in the past (see e.g. Davis, 1984; Braswell et al., 1997), and it has been adopted in many studies more recently (Anderson et al., 2010; Kuzyakov and Gavrichkova, 2010; Chen et al., 2014; Rammig et al., 2014). These studies indicate that





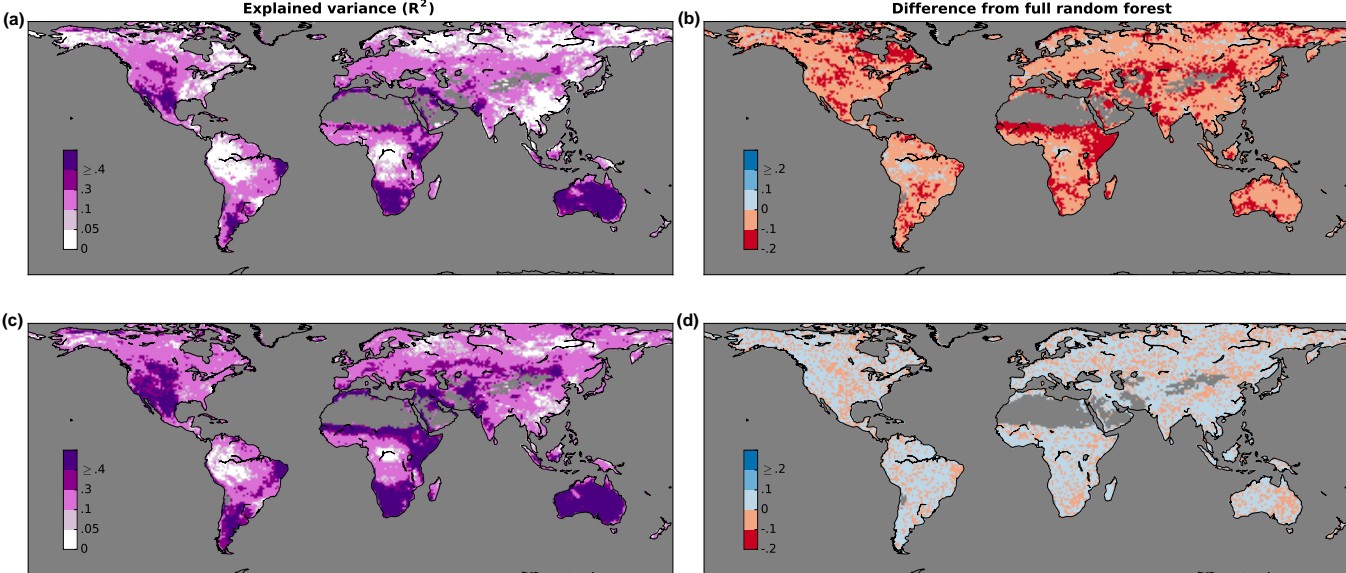

**Figure 6.** Analysis of spatiotemporal aspects of our framework. **(a)** Explained variance ($R^2$) of NDVI anomalies based on a full random forest model in which all climatic variables are included as predictors and in Fig. 5a, except for the past cumulative climate and climate extreme indices (see Sect. 3.3). **(b)** Difference in terms of $R^2$ between the model without cumulative and extreme predictors and the full random forest model in Fig. 5a. **(c)** Explained variance ($R^2$) of NDVI anomalies based on a full random forest model in which all climatic variables are included as predictors and in Fig. 5a, but including also the predictors from the 8 nearest neighbors. **(d)** Difference in terms of $R^2$ between this full random forest model which includes spatial information from neighbouring pixels and the full random forest model in Fig. 5a.

lag times depend on both the specific climatic control variable and the characteristics of the ecosystem. As explained in Sect. 3.3, in our analysis shown in Fig. 4 and 5 we moved beyond traditional cross-correlations, and incorporate higher-lever variables in the form of cumulative and lagged responses to extreme climate. As mentioned in Sect. 3.3, our experiments indicated that lags of more than six months do not add extra predictive power (not shown), even though the effect of anomalies in water

5   availability on vegetation can extend for several months (Papagiannopoulou et al., *in review*).

To disentangle the response of vegetation to past cumulative climate anomalies and climatic extremes, Fig. 6a visualizes the predictive performance when cumulative variables and extreme indices are not included as predictive variables in the random forest model. As shown in Fig. 6b, in almost all regions of the world the predictive performance decreases substantially compared to the full random forest model approach, i.e. using the full repository of predictors, (Fig. 5a), especially in the

10  Sahel, the Horn of Africa or North America. In those regions 10-20% of the variability in NDVI is explained by the occurrence of prolonged anomalies and/or extremes in climate, illustrating again the non-linear responses of vegetation to climate. For



more detailed results about lagged vegetation responses for specific climate drivers and the effect of climate extremes on vegetation, the reader is referred to Papagiannopoulou et al. (*in review*).

Because of uncertainties in the observational records used in our study to represent climate and predict vegetation dynamics, and given that ecosystems and regional climate conditions usually extend over areas that exceed the spatial resolution of these

records, one may expect the predictive performance of our models become more robust when including climate information from neighboring pixels.

We therefore also consider an extension of our framework to exploit spatial autocorrelations, inspired by Lozano et al. (2009), who achieved spatial smoothness via an additional penalty term that punishes dissimilarity between coefficients for spatial neighbors. In our analysis, we incorporate at a given pixel spatial autocorrelations by extending the predictor variables

of our models with the predictor variables of the 8 neighboring pixels. We provide such an extension both for the full and the baseline random forest model. As such, for the full random forest model, a vector of 34,956 (3,884 × 9) predictor variables is formed for each pixel.

Figure 6c illustrates the performance of the full random forest model that includes the spatial information. As one can observe in Fig. 6d, the explained variance of NDVI anomalies remains similar to the original model which depicts the same

approach without spatial autocorrelation (Fig. 5a). While in most areas the performance slightly increases, the explained variance never improves by more than 10%; as a result, incorporating spatial autocorrelations in our framework does not seem to further improve the quantification of Granger causality and is not considered in further applications of the framework (see Papagiannopoulou et al., *in review*).

### 5.2 The importance of focusing on vegetation anomalies

In Sect. 3.2 we advocated that Granger causality analysis should target on NDVI anomalies, as opposed to raw NDVI values. There are several fundamental reasons. First, by applying a decomposition, one can subtract long-term trends from the NDVI time series, making the resulting time series more stationary. This is absolutely needed, as existing Granger causality tests cannot be applied for non-stationary time series. Secondly, by subtracting the seasonal cycle from the time series, one is able not only to remove a confounding factor that may contribute predictive power without having causality, but also to remove a

clear autoregressive component that can be well explained from the NDVI time series themselves. As vegetation has a strong seasonal cycle, it is not difficult to predict subsequent vegetation conditions by using the past observations of the seasonal cycle only. To corroborate this aspect, we repeat our analysis in Sect. 4.2, but this time considering the raw NDVI time series instead of the NDVI anomalies are considered as the target variable. We again compare the full and the baseline random forest models.

The results are visualized in Fig. 7a. As it can be observed, worldwide the $R^2$ is close to the optimum of one. However, due

to the overwhelming domination of the seasonal cycle, it becomes very difficult, or even impossible, to unravel any potential Granger-causal relationships with climate time series in the northern hemisphere – see Fig. 7b. The predictability of NDVI based on the seasonal NDVI cycle itself is already so high that nothing can be gained by adding additional climatic predictor variables; see also the large amplitude of the seasonal cycle of NDVI at those latitudes compared to the NDVI anomalies, as illustrated in Fig. 2. Therefore, a non-linear baseline autoregressive model is able to explain most of the variance in the time



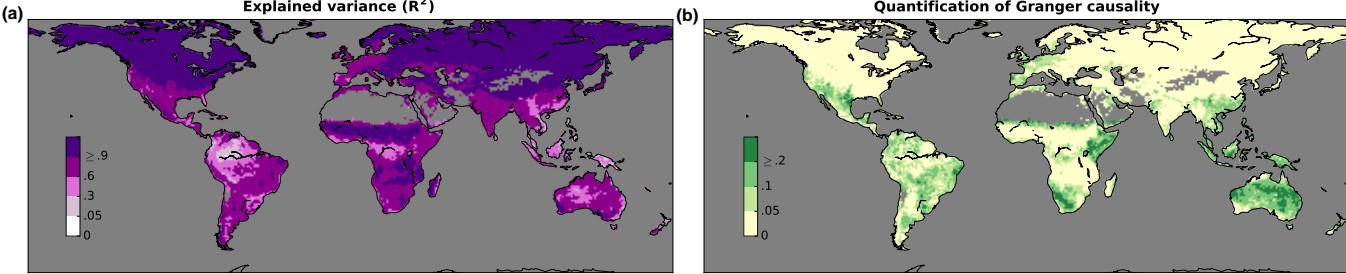

**Figure 7.** Comparison of model performance with $R^2$ as metric with the raw NDVI time series as target variable. **(a)** Full random forest model **(b)** Improvement in terms of $R^2$ of the full random forest model over the baseline random forest model.

series. Moreover, as observed in Fig. 1, temperature and radiation also manifest strong seasonal cycles that coincide with the NDVI cycle. For most regions on Earth, such a stationary seasonal cycle is less present for variables such as precipitation. This can potentially yield wrong conclusions. For instance, temperature in the northern hemisphere becomes really predictive for raw NDVI, since the two seasonal cycles have the same pattern. However, based on the above discussion, it is obvious that

results of that kind should be treated with great caution. The sudden rise in importance for temperature can simply be explained by the presence of a stronger seasonal cycle for this variable. We therefore conclude that, for climate data, a Granger causality analysis should be applied after decomposing time series into seasonal anomalies.

## 6   Conclusions

In this paper we introduced a novel framework for studying Granger causality in climate-vegetation dynamics. We compiled

a global database of observational records spanning a thirty-year time frame, containing satellite, *in situ* and reanalysis-based datasets. Our approach consists of the combination of data fusion, feature construction and non-linear predictive modelling on climate and vegetation data. The choice of random forest as a non-linear algorithm has been motivated by its excellent computational scalability with regards to extremely large data sets, but could be easily replaced by any other non-linear machine learning technique, such as neural networks or kernel methods.

Our results highlight the non-linear nature of climate–vegetation interactions and the need to move beyond the traditional application of Granger causality within a linear framework. Comparisons to traditional Granger causality approaches based indicate that our random forest framework can predict 14% more variability of vegetation anomalies on average globally. The predictive power of the model is especially high in water-limited regions where a large part of the vegetation dynamics responds to the occurrence of antecedent rainfall. Moreover, our results indicate the need to consider multi-month antecedent

periods to capture the effect of climate on vegetation, and the need for considering the impact of climate extremes on vegetation





dynamics. The reader is referred to Papagiannopoulou et al. (*in review*) for a detailed analysis of the effect of different climate predictors on the variability of global vegetation using the mathematical approach described here.

## 7   Code and data availability

We use the implementation of scikit-learn (Pedregosa et al., 2011) library in Python for the random forest regressor. Data used
5   in this manuscript can be accessed using http://www.SAT-EX.ugent.be as gateway.

*Author contributions.*  Diego G. Miralles, Willem Waegemann and Niko E.C. Verhoest conceived the study. Christina Papagiannopoulou conducted the analysis. Willem Waegeman, Diego G. Miralles and Christina Papagiannopoulou led the writing. All co-authors contributed to the design of the experiments, discussion and interpretation of results, and editing of the manuscript.

*Acknowledgements.*  This work is funded by the Belgian Science Policy Office (BELSPO) in the framework of the STEREO III programme,
10   project SAT-EX (SR/00/306). D. G. Miralles acknowledges financial support from The Netherlands Organization for Scientific Research through grant 863.14.004. W. Dorigo is supported by the 'TU Wien Wissenschaftspreis 2015', a personal grant awarded by the Vienna University of Technology. The authors thank Mathieu Depoorter, Stijn Decubber and Julia Green for the fruitful discussions. Finally, the authors sincerely thank the individual developers of the wide range of global data sets used in this study.



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
