# Peer review of "A non-linear Granger causality framework to investigate climate-vegetation dynamics"

_Geoscientific Model Development, 2016_

## Referee Comment (RC1) · Anonymous Referee #1 · 30 Jan 2017

———————————————

A non-linear Granger causality framework to investigate climate-vegetation dynamics

by

Christina Papagiannopoulou, Diego G. Miralles, Niko E. C. Verhoest, Wouter A. Dorigo, and Willem Waegeman

———————————————

The manuscript introduces a Granger causal inference approach to investigate climate-vegetation dynamics. A great effort in collecting a representative enough dataset has

been pursued to study such dependencies. The authors put emphasis on the non-linearity of the approach since the VAR method typically used in the canonical Granger approach is here replaced by a nonlinear regression tool, the random forests method. Authors claim that the causal patterns are more clearly identifiable than with traditional linear models. Overall, I think this is a very nice piece of work that worths publishing after some clarifications and addressing some problems. Below authors will find a long list of minor and major comments that I hope they can address.

- abstract:

3: unravel the influence... : this looks like an ambitious goal that I'm not sure authors finally managed to address 4: existing statistical methods: do authors refer to linear ones only, right? 8: (also in the title) the word 'framework' looks too ambitious. In the end, authors only proposed to follow the Granger approach with a different feature selection and regression method. Does this qualify to call it framework?

p2.29: y alludes to the NDVI time series: shouldn't be the IAV of NDVI thereof?

p3.7: for me, describing the $R^2$ is too verbose and useless in a scientific journal nowadays

p3.eq2-3: the \approx symbol is meaningless here. I'd suggest to include the signal model here ($y = \hat y + e$), and describe the assumptions about the noise model (Gaussian, uncorrelated?). Also, I don't find natural that both eqs. have the same model coefficients \beta_11p.

p3.27: authors should clarify the sentence "neither variables nor observational ... and errors are ...". Independent of what? each other? independent noise? Please be explicit and consistent in the use of the terms 'error', 'noise', 'residuals'.

p4.10, eq4: describe the meaning of \beta_13 and all terms involved in the equation

p4.26: Maybe I'm missing something but if you split the data this way, aren't you discarding long-term correlations. Also, by simple xval, results depend to a large extent of

the selected data splits. To avoid this, why not LOO?

p5.10: the same comment about the \approx symbol before: please include the signal model equations here too.

p5.15: formally it is straightforward, but not computationally or for decision making which may be an infeasible problem.

p6.1-3: if you want to keep this statement, please discuss about the theoretical implications, and cite other nonlinear Granger causality methods (a simple search in Google will return you several dozens of works in machine learning, kernel methods, time series forecasting, econometrics and finance).

p6.1-14: verbose, remove or summarize a lot.

p9.eq: the upperscript T may confuse as in standard algebra that symbol stands for transpose.

p9.3: obvious non-stationary: sometimes it is not that obvious.

p12.6: a sentence does not conform a paragraph. And by the way... is $1°$ enough resolution to claim something about causation? do the expected relations occur at such broad scale?

p12.13: please avoid overoptimistic phrases like "our nonlinear random forestS".

p12.17: "simple correlations" should be "spurious correlations"? in any case this sentences deserves more clarification and be more explicit

Fig4: some discussions and words of caution should be given about deriving conclusions out of R2~0.4. By the way, why the maximum in the scale is not explicit for R2 and you select that threshold in 0.4? Why not using the statistical significance of the correlation rather than the $R^2$ score? Can authors include and discuss the maps of R p-values?

Fig4 caption: 'with respect to a the' to be corrected

p13.3: 'our'?

p14.3: what are these patterns of the explained variance? some clarification is needed here? I guess authors refer to spatial patterns of variation? If that is the case, it looks not really obvious to talk about spatial relations when no such relations are considered to build up the regression models.

p14.7: unambiguous? some more comments are needed, and if possible supported by numerical scores.

p13-14: as a reader I'd prefer to have in the same figure panel the current figures 4 and 5 so I could directly compare results in one shot.

p15.3: what do authors mean by 'higher-lever variables'? are you thinking of higher-order statistical relations between variables? this is absolutely confusing.

p15.5: please provide a copy of the (Papagiannopolou et al, in review) so reviewers can appreciate differences in approaches and results. Alternative, cite an accessible work to support the claims in this paper.

p16.3-18: please clarify these paragraphs in several ways: 1) the spatial encoding is not at all clear since typically the input (feature) space is augmented with the neighbors which are then used to predict on the central pixel (the length of the observation variable does not change), which seems not to be the case here. 2) it is weird that the spatial info didn't improve the results: I'd thank the authors to include such 'negative results' but then some comments and clarifications are needed (e.g. 1° is already integrating too much info, or spatial encoding was not taking into account pixel spatio-temporal variances?)

p17.9: as said before I feel claiming a 'novel framework' is far too much for this contribution.

p17.15-20: some claims are contained here without empirical justification. I think that authors lost a nice opportunity here to explain the causal relations. For example, to me it seems ad hoc to justify results with a simple 'the predictive power of the model is especially high in water-limited regions'. Probably this is true but some numbers are needed to support it. I suggest to include a summarizing feature ranking of the LR vs RFs (e.g. permutation analysis, and surrogate analysis). Also, summarize results per regions and biomes would help discussing the results more profoundly, elevating the debate. Of course, these two issues may require some more work, but I sincerely think they are mandatory to make a sound publication.

p18.8: reproducibility is not possible as data is not available yet. do authors plan to make these data available to the community?

---

## Author Comment (AC1) · 3 Feb 2017

**Response to Anonymous Referee #1**

The manuscript introduces a Granger causal inference approach to investigate climate-vegetation dynamics. A great effort in collecting a representative enough dataset has been pursued to study such dependencies. The authors put emphasis on the non-linearity of the approach since the VAR method typically used in the canonical Granger approach is here replaced by a nonlinear regression tool, the random forests method. Authors claim that the causal patterns are more clearly identifiable than with traditional linear models. Overall, I think this

**is a very nice piece of work that is worth publishing after some clarifications and addressing some problems.**

We would like to thank the reviewer for his/her appreciation of the manuscript, for the constructive feedback, and the thorough assessment of the manuscript. Below we provide a point-to-point response to each comment.

**Below authors will find a long list of minor and major comments that I hope they can address.**
**- abstract: 3: unravel the influence... : this looks like an ambitious goal that I'm not sure authors finally managed to address**

The sentence reads 'Data of this kind [Earth observations] provide new means to unravel the influence of climate on vegetation dynamics'. We are sure the reviewer would agree that they do, so we do not really see an issue regarding this sentence.

**- 4: existing statistical methods: do authors refer to linear ones only, right?**

'Existing' will be replaced by 'commonly used'. We believe this addresses the issue the reviewer refers to.

**- 8: (also in the title) the word 'framework' looks too ambitious. In the end, authors only proposed to follow the Granger approach with a different feature selection and regression method. Does this qualify to call it framework?**

We understand the concern, yet 'framework' can be defined as 'a basic conceptional structure (as of ideas)' (Merrian-Webster dictionary). We chose to use the word 'framework', because we believe it reflects well the conceptional structure followed here, which consists of several sequential steps. First, multiple datasets of the

most important climatic variables have been collected and converted into a common temporal and spatial resolution (a multidimensional data-cube). Second, by applying feature extraction techniques and domain knowledge, predictor variables have been constructed. Third, a nonlinear machine learning algorithm has been designed and applied. Forth, causality has been assessed based on Granger causality tests. As such, we think that we propose a complete framework that can be used for knowledge discovery in climate sciences, and which in this paper is applied to unveil climate-vegetation dynamics, but that could be used to detect other causal patterns in the climate system. Moreover, by using nonlinear models and feature extraction techniques, we argue that this framework substantially differs from methodologies that are common practice in the field.

**p2.29: y alludes to the NDVI time series: shouldn't be the IAV of NDVI thereof?**

Yes, it is true that we finally model the IAV of NDVI, this is just a starting point of explaining the basic model. We will clarify it in the revised manuscript.

**p3.7: for me, describing the $R^2$ is too verbose and useless in a scientific journal nowadays**

The $R^2$ is indeed a well-known performance measure, we wrote down the formula just to avoid confusion: for linear models one often computes the $R^2$ as a correlation coefficient, while for nonlinear models one has to compute the $R^2$ in a different way, using the formula in the manuscript. This is perhaps obvious, but it might be useful to have the formula in the paper for readers that are less familiar with the different definitions of $R^2$.

**p3.eq2-3: the $\approx$ symbol is meaningless here. I'd suggest to include the**

[Figure]

**signal model here (**$y = \hat{y} + e$**), and describe the assumptions about the noise model (Gaussian, uncorrelated?). Also, I don't find natural that both eqs. have the same model coefficients** $\beta_{11p}$**.**

We agree. We will incorporate these changes.

**p3.27: authors should clarify the sentence "neither variables nor observational ... and errors are ...". Independent of what? each other? independent noise? Please be explicit and consistent in the use of the terms 'error', 'noise', 'residuals'.**

We meant independent from each other. We will clarify this in the revised version.

**p4.10, eq4: describe the meaning of** $\beta_{13}$ **and all terms involved in the equation**

We believed this was already clear, but perhaps it is not the case; we will include a more extensive explanation of the tri-variate extension of Granger causality in the revised manuscript. In the tri-variate case (as in the case of the bivariate Granger causality) we examine if the time series x Granger-causes the time series y. The time series w might act as a confounder in deciding whether x Granger-causes y; for this reason it is included in both models (baseline and full). That way the method can cope with cross-correlations between climatic drivers of vegetation. The $\beta_{13}$ are the coefficients of the time series w while the rest of the coefficients have the same meaning as in the bivariate case.

**p4.26: Maybe I'm missing something but if you split the data this way, aren't you discarding long-term correlations. Also, by simple xval, results depend to a large extent of the selected data splits. To avoid this, why not LOO?**

[Figure]

We are not sure whether we understand this comment. Our motivation for doing 5-fold cross-validation instead of leave-one-out was mainly motivated by computational reasons. LOO takes a long time to compute and is generally not the recommended method when analyzing large datasets. As we are working with an extremely large dataset here, computational efficiency is always the first criterion to look for.

**p5.10: the same comment about the $\approx$ symbol before: please include the signal model equations here too.**

Thanks. We will revise this as well.

**p5.15: formally it is straightforward, but not computationally or for decision making which may be an infeasible problem.**

Of course. We only want to convey here that the formal definition of Granger causality does not change in the case of more than three time series.

**p6.1-3: if you want to keep this statement, please discuss about the theoretical implications, and cite other nonlinear Granger causality methods (a simple search in Google will return you several dozens of works in machine learning, kernel methods, time series forecasting, econometrics and finance).**

We agree that there are previous works on nonlinear Granger causality. Those methods typically assume stationarity of the time series, and they are hence not immediately applicable for climatic time series. We will extend the paragraph with a more thorough discussion of related work and new references to these articles. We should also clarify that we have not introduced nonlinear Granger causality. But these

more complex methods which use Granger causality have not been widely applied in the field of climate sciences.

**p6.1-14: verbose, remove or summarize a lot.**

This might be obvious for a well-informed reader, but we believe that an explanation of that kind is needed for readers that are less familiar with Granger causality and time series forecasting. We will nonetheless try to condense these sentences without losing information.

**p9.eq: the upperscript T may confuse as in standard algebra that symbol stands for transpose.**

We will replace the symbol T with Tr in order to make it clearer in the revised manuscript.

**p9.3: obvious non-stationary: sometimes it is not that obvious.**

We propose to delete the word 'obvious', indeed.

**p12.6: a sentence does not conform a paragraph.**

We disagree, a paragraph is 'a subdivision of a written composition that consists of one or more sentences' (Merrian-Webster dictionary, but also any other...). Yet, we will consider merging it with the previous paragraph, although the decision to make it a stand-alone sentence was to highlight it.

**And by the way... is 1 degree enough resolution to claim something about causation? do the expected relations occur at such broad scale?**

[Figure]

Most atmospheric variables change consistently at spatial resolutions that are even coarser than 1 degree; in fact current climate models resolve the land–atmospheric interactions while working at coarser resolutions. Nonetheless, in heterogeneous environments this would be a limitation, and this issue should be acknowledged in the revised version of the manuscript. We also note that there is a trade-off between spatial resolution and time span. The 1-degree resolution is a characteristic of the datasets we are working with, and if we wanted to focus on finer resolutions, we would need to incorporate datasets from sensors covering more recent years only. The 1-degree resolution, in addition, still also allows us to perform our calculations in a reasonable amount of time.

**p12.13: please avoid overoptimistic phrases like "our nonlinear random forestS".**

Thanks, this will be rephrased to "the nonlinear random forests models".

**p12.17: "simple correlations" should be "spurious correlations"? in any case this sentences deserves more clarification and be more explicit Fig4: some discussions and words of caution should be given about deriving conclusions out of $R^2 \sim 0.4$. By the way, why the maximum in the scale is not explicit for $R^2$ and you select that threshold in 0.4? Why not using the statistical significance of the correlation rather than the $R^2$ score? Can authors include and discuss the maps of R p-values?**

This is a remark that we expected. As mentioned in the manuscript, the assumptions of common statistical tests are violated due to the non-stationarity of the data and the nonlinearity of the proposed model. Developing a statistical test that is able to handle non-stationary time series and nonlinear models is not a trivial task.

As far as we know, no such test exists. Therefore, we decided to focus on expressing Granger causality in a quantitative way instead of a qualitative way, and stress the gained improvement with the use of a nonlinear model. We are currently developing a statistical testing procedure, but this is work in progress and will be the subject of future contributions. We will also include the relevant references and a more thorough discussion about existing statistical tests in the revised manuscript.

**Fig4 caption: 'with respect to a the' to be corrected**

Thanks.

**p13.3: 'our'?**

Yes, thanks.

**p14.3: what are these patterns of the explained variance? some clarification is needed here? I guess authors refer to spatial patterns of variation? If that is the case, it looks not really obvious to talk about spatial relations when no such relations are considered to build up the regression models.**

In fact this section does not refer to spatial patterns, but to a general improvement of the full model versus the restricted model. We will update the text accordingly to clarify this.

**p14.7: unambiguous? some more comments are needed, and if possible supported by numerical scores.**

With unambiguous we just mean that the improvement is clearly visible here (in the order of 20 to 60%). This claim is supported by Figure 5b.
**p13-14: as a reader I'd prefer to have in the same figure panel the current figures 4 and 5 so I could directly compare results in one shot.**

We understand this concern and understand that a 3 × 2 figure would be more convenient for the reader. However, we chose to have Figure 5 in a separate panel because this is the main figure of the paper, and unless the reviewer is insisting on this issue we would rather keep it this way.

**p15.3: what do authors mean by 'higher-lever variables'? are you thinking of higher-order statistical relations between variables? this is absolutely confusing.**

With the term 'higher-level variables' we refer to past cumulative climate and climate extreme indices which are in the dataset as predictor variables. We will clarify this in the revised manuscript.

**p15.5: please provide a copy of the (Papagiannopolou et al, in review) so reviewers can appreciate differences in approaches and results. Alternative, cite an accessible work to support the claims in this paper.**

The referred article will be enclosed to the resubmission of the revised paper, as long as it can be made available to the editorial and reviewers only.

**p16.3-18: please clarify these paragraphs in several ways: 1) the spatial encoding is not at all clear since typically the input (feature) space is augmented with the neighbors which are then used to predict on the central pixel (the length of the observation variable does not change), which seems not to be the case here. 2) it is weird that the spatial info didn't improve the results: I'd**

**thank the authors to include such 'negative results' but then some comments and clarifications are needed (e.g. 1 degree is already integrating too much info, or spatial encoding was not taking into account pixel spatiotemporal variances?)**

Yes, the approach we followed is as described by the reviewer. The feature space of one pixel is augmented with the features of the 8 neighboring pixels. We also expected to see a substantial improvement using spatial information but this was not the case. We think that this is due to the huge dimension of the feature space which may include redundant information, in combination with the low number of observations per pixel. We will extend the discussion in that direction. Let us also stress that we are currently working on (more complex) spatial models that could overcome this kind of issues.

**p17.9: as said before I feel claiming a 'novel framework' is far too much for this contribution.**

See above response.

**p17.15-20: some claims are contained here without empirical justification. I think that authors lost a nice opportunity here to explain the causal relations. For example, to me it seems ad hoc to justify results with a simple 'the predictive power of the model is especially high in water-limited regions'. Probably this is true but some numbers are needed to support it. I suggest to include a summarizing feature ranking of the LR vs RFs (e.g. permutation analysis, and surrogate analysis). Also, summarize results per regions and biomes would help discussing the results more profoundly, elevating the debate. Of course, these two issues may require some more work, but I sincerely think they are mandatory to make a sound publication.**

Actually, we have performed this kind of analyses, taking feature rankings using RFs. However, it has been shown that those rankings become unstable due to highly-correlated predictors. A specialized approach would be needed here, in which groups of features are ranked instead of individual features. This makes the rankings more stable and improves the interpretability. It is exactly what we do in the complementary paper (Papagiannopoulou et al., under review).

We agree with the reviewer that a stratification of the results according to regions/biomes is a relevant addition to the paper. The revised version will provide the results stratified according to IGBP vegetation classes for both the baseline and full random forest model. These new results will be discussed in the relevant sections.

**p18.8: reproducibility is not possible as data is not available yet. do authors plan to make these data available to the community?**

All codes will be freely available and documented on GitHub and will comply with the Copernicus data policy. On the other hand, the database is formed by a collection of datasets that are all publicly available and that, due to copyright conflicts, we cannot openly distribute. Nonetheless, we have decided to add the link to these datasets below Table 1 in the revised version.

---

## Referee Comment (RC2) · Anonymous Referee #2 · 21 Feb 2017

Title: A non-linear Granger causality framework to investigate climate–vegetation dynamics

Authors: Christina Papagiannopoulou, Diego G. Miralles, Niko E. C. Verhoest, Wouter A. Dorigo, and Willem Waegeman

General Comments: Reviewer summary: The manuscript presents a non-linear Granger causality analysis to investigate climate-vegetation interactions. Anomalies of the normalized vegetation index (NDVI) are analyzed in conjunction with a full set of climate variables taken from re-analysis, in situ, and satellite observations. The data provide multi-decadal global coverage for water availability (precipitation, snow water equivalent and soil moisture data), temperature, and radiation. All data spans the

period 1981-2010 at the global scale and has been converted to a common monthly temporal resolution and 1x1 degree spatial resolution. At each pixel the NDVI data is considered the response and the climate data the predictor variables. A moving window of twelve months is used to determine if the climate data granger-causes the NDVI value. Analysis is performed on NDVI anomalies computed by subtracting the corresponding monthly expectation from the de-trended time series. The climate data as well as cumulative values and extreme indices calculated from the climate data were included as predictor variables. The non-linear Granger causality uses a non-linear random forest model, and is shown to explain more of the variance than the linear granger analysis.

Article contribution and overall impact: This study makes an effort to use multiple climate data sources to tease out predictability for vegetation anomalies. The authors highlight improvements with the non-linear method compared to traditional granger causality, as well as the importance of using extreme events. The discussion would benefit from a more explicit discussion of the uncertainty associated with the climate datasets used as predictors. Given that this study precedes or supports Papagiannopoulou et al (in review), more discussion of those results and their importance would be useful as that study is not available to the reader. Specifically, the follow-on study highlights the importance of specific climate predictors for particular regions. It is not clear how those variables are chosen from the many climate predictors, and it would be useful to provide an example in this manuscript to highlight the strength of this method with a clear detailed regional example.

Detailed comments:

Page 1 line 17-18: Should this read "predictions of vegetation in response to future climate can be improved through a better understanding…" ? as you are looking for climate drivers of vegetation.

Page 2 line 22: define "higher-level features" here and throughout manuscript. It is not

clear what these are. (Pg 11 line 4, pg.15 line 2)

Page 2 line 24: define "higher-level climate variables" not clear what this is.

Page 3 line 2-7: May not be necessary to include full definition of R2.

Page 3 line 30: update "might lead to wrong" to "might lead to incorrect"

Page 12 line 15-23: Are the results for all variables, or the most predictive variable, or a set of variables at each pixel?

Page 12 line 26-27: Why is this chosen as the minimum? Please explain or provide citation.

Page 13 line 10: by what margin is the uncertainty larger in these regions, and for what reasons? Are you referring to all the climate variables, if not please qualify. The citation references error for soil moisture. Add citations, which support the amount of uncertainty in these regions for the remaining data types.

Page 13 line 7 to bottom and page 14 line 1-4: Move this to discussion.

Page 14 line 1: Update to "vegetation anomalies are not necessarily"

Page 14 line 7: Use different phrasing for "unambiguous"

Page 14 line 7-12: move to discussion.

Page 14 line 8-10: Recommend re-wording this. The limit for figure 5 and the presentation of the non-linear analysis is still to a limit of R2 = 0.4 as in figure 4? An R2 of 0.4 does not seem like a strong correlation. Though figure 5 is improved from figure 4 there are large portions that show no improvement, and the overall explained variance is below 40% in most regions.

Page 14 line 10: "comparison between figs 4b and 5b" explain in more detail. It would be easier for the reader to compare these if they were in one figure block, or on the same page.

Page 15 line 5: Please provide more detail about this study. It comes up frequently in the manuscript, and a larger summary with details (supportive numbers or examples from regions) would be helpful since we do not have access to the manuscript.

Page 15 line 11: Has a test been run with only the anomalies and extremes? Would that sub-set of predictors provide strong predictive performance?

Page 16 line 1-2: Provide more detail from supporting manuscript for current manuscript. It is necessary to support this analysis that you can separate specific drivers.

Page 16 line 3-6: Connect this sentence to the following paragraph.

Page 16 line 17: Is the "framework" the non-linear component? Maybe just call it that – non-linear, rather than a framework. This implies a more complex process.

Page 17 line 11: explain "feature construction"

Page 17 line 16: update word order to read "causality based approaches indicate"

———————————————

---

## Author Comment (AC2) · 28 Feb 2017

**Response to Anonymous Referee #2**

**General Comments:**
**Reviewer summary: The manuscript presents a non-linear Granger causality analysis to investigate climate-vegetation interactions. Anomalies of the normalized vegetation index (NDVI) are analyzed in conjunction with a full set of climate variables taken from re-analysis, in situ, and satellite observations. The data provide multi-decadal global coverage for water availability (precipitation, snow water equivalent and soil moisture data), temperature, and radiation. All**

data spans the period 1981-2010 at the global scale and has been converted to a common monthly temporal resolution and 1×1 degree spatial resolution. At each pixel the NDVI data is considered the response and the climate data the predictor variables. A moving window of twelve months is used to determine if the climate data Granger-causes the NDVI value. Analysis is performed on NDVI anomalies computed by subtracting the corresponding monthly expectation from the de-trended time series. The climate data as well as cumulative values and extreme indices calculated from the climate data were included as predictor variables. The non-linear Granger causality uses a non-linear random forest model, and is shown to explain more of the variance than the linear Granger analysis.

Article contribution and overall impact: This study makes an effort to use multiple climate data sources to tease out predictability for vegetation anomalies. The authors highlight improvements with the non-linear method compared to traditional Granger causality, as well as the importance of using extreme events. The discussion would benefit from a more explicit discussion of the uncertainty associated with the climate datasets used as predictors. Given that this study precedes or supports Papagiannopoulou et al (in review), more discussion of those results and their importance would be useful as that study is not available to the reader. Specifically, the follow-on study highlights the importance of specific climate predictors for particular regions. It is not clear how those variables are chosen from the many climate predictors, and it would be useful to provide an example in this manuscript to highlight the strength of this method with a clear detailed regional example.

We would like to thank the reviewer for the feedback, and the thorough assessment of the manuscript.

We agree that the study Papagiannopoulou et al. (in review), in which we apply the method to discern the importance of different climatic drivers, may be useful for the

referees to assess the potential of our framework. As we mentioned in our response to the Anonymous Referee #1, that article will be enclosed in the resubmission of the revised paper, so it can be available to the editor and reviewers. As the referred article is a follow on from this GMD paper, as the referee states, we do not see the need to provide details about its specific results within the GMD paper.

Regarding the above comment on input uncertainty, the revised article will include a few statements discussing the impact of these uncertainties.

Below we provide our pointwise response to the reviewer.

**Detailed comments:**
**Page 1 line 17-18: Should this read "predictions of vegetation in response to future climate can be improved through a better understanding..." ? as you are looking for climate drivers of vegetation.**

We think that the initial sentence "Because of the strong two-way relationship between terrestrial vegetation and climate variability, predictions of future climate can be improved through a better understanding..." is in fact correct. In this paragraph, we discuss the complex two-way interactive relationship between vegetation and climate in order to state the importance of understanding climate dynamics to predict climate accurately. Therefore, a better understanding of the vegetation response to past climate variability, brings us one step further in understanding future climate, since the latter will also be affected by the fate of vegetation.

**Page 2 line 22: define "higher-level features" here and throughout manuscript. It is not clear what these are. (Pg 11 line 4, pg.15 line 2)**

With the terms 'higher-level features' or 'higher-level climate variables' we refer to the 'cumulative' and 'lagged variables' as well the climate extreme indices, which have been calculated on the raw time series and serve as additional predictor variables

**[GMDD](GMDD)**

Interactive
comment

in the dataset. We will clarify this in the revised manuscript.

**Page 2 line 24: define "higher-level climate variables" not clear what this is.**

True. See above response.

**Page 3 line 2-7: May not be necessary to include full definition of $R^2$.**

This comment has been addressed in our response to Anonymous Referee #1. We acknowledge the reviewers' claim is correct, but we should clarify the different definitions of $R^2$ in linear and non-linear models. For non-linear models, $R^2$ is calculated using the formula in the manuscript, while for linear models it can be calculated as a correlation coefficient. We believe that since the audience of GMD is rather multidisciplinary, including this definition may be helpful to some readers.

**Page 3 line 30: update "might lead to wrong" to "might lead to incorrect"**

Yes, we will change this phrase in the revised manuscript.

**Page 12 line 15-23: Are the results for all variables, or the most predictive variable, or a set of variables at each pixel?**

We use all the variables at each pixel in order to obtain the results presented in this section. We will clarify it in the revised manuscript.

**Page 12 line 26-27: Why is this chosen as the minimum? Please explain or provide citation.**

This statement comes from the definition of Granger causality. The minimum explained variance can be achieved by using the history of the target variable only, and this is basically the model referred to as 'baseline model' in the manuscript. We will make this more explicit in the revised version.

**Page 13 line 10: by what margin is the uncertainty larger in these regions, and for what reasons?**

As one can notice from the map in Figure 5b, the improvement of the full model in terms of $R^2$ compared to the baseline is low. Therefore, the results indicate that the Granger causal effects of climate on vegetation anomalies in these regions are not obvious. This is why we enumerate a set of studies which explore the main drivers of vegetation in these regions, explaining the poor predictive performance of the full model with respect to the baseline model. We will extend the discussion of this part in the revised manuscript to clarify this point.

**- Are you referring to all the climate variables, if not please qualify.**

Yes, we are referring to all the climate variables included in the dataset. We will clarify it in the revised manuscript.

**- The citation references error for soil moisture.**

Thanks, we will update.

**- Add citations, which support the amount of uncertainty in these regions for the remaining data types.**

See above response.

**Page 13 line 7 to bottom and page 14 line 1-4: Move this to discussion.**

The subject of this section is to present the results obtained from the proposed methodology. The part the reviewer refers to discusses the performance of the proposed methodology in comparison to other models. We therefore believe that this is the appropriate location to place this discussion, and we would prefer to keep it this way, unless the reviewer is insisting on this issue.

**Page 14 line 1: Update to "vegetation anomalies are not necessarily"**

We will add the word 'anomalies' in the revised manuscript.

**Page 14 line 7: Use different phrasing for "unambiguous"**

By "unambiguous" we just mean that the improvement is clearly visible here, which we believe reflects correctly the meaning of this word.

**Page 14 line 7-12: move to discussion.**

See the above response.

**Page 14 line 8-10: Recommend re-wording this. The limit for figure 5 and the presentation of the non-linear analysis is still to a limit of $R^2$ = 0.4 as in figure 4? An $R^2$ of 0.4 does not seem like a strong correlation. Though figure 5 is improved from figure 4 there are large portions that show no improvement, and the overall explained variance is below 40% in most regions.**

We kept the same colorbar scale in both figures for a better figure-to-figure comparison. We note that $R^2$ is not to be mistaken by 'correlation' in the non-linear models; it is a performance measure that indicates how close the model predictions are to the real values of the target variable. In our study, we remove the seasonal cycle from the NDVI time series and we target the NDVI anomalies, making the task more difficult than predicting the raw NDVI time series, since the autocorrelation in the NDVI anomalies time series is much lower. Note that if we target the raw NDVI time series (which includes the seasonal component), the $R^2$ is close to 1 in most of the regions (see Fig. 7 in the manuscript). In addition, it is worth noting that there are other factors such as fires or harvest that affect vegetation dynamics but are not included in the dataset, as mentioned in the discussions. Therefore, we should be aware that we focus on explaining the variance of the NDVI anomalies, taking into account only climatic variables, and focusing on the part of their explanatory power that truly reflects Granger-causal relationships.

**Page 14 line 10: "comparison between figs 4b and 5b" explain in more detail. It would be easier for the reader to compare these if they were in one figure block, or on the same page.**

Yes, it is true that a 3 × 2 figure would be more convenient for the reader; it was also a comment from the Anonymous Referee #1. However, Figure 5 is the main figure of the paper and we decided to make it in a separate block to highlight it.

**Page 15 line 5: Please provide more detail about this study. It comes up frequently in the manuscript, and a larger summary with details (supportive numbers or examples from regions) would be helpful since we do not have access to the manuscript.**

See first response.
**Page 15 line 11: Has a test been run with only the anomalies and extremes? Would that sub-set of predictors provide strong predictive performance?**

We agree with the reviewer. We can include more experimental results at this point in order to figure out the importance of anomalies and extremes in particular. In the paper Papagiannopoulou et al. (in review), there is a thorough discussion about the different groups of variables that have been tested for Granger causality, and provides the results of isolating the impact of extremes. Again this paper will be enclosed in the resubmission.

**Page 16 line 1-2: Provide more detail from supporting manuscript for current manuscript. It is necessary to support this analysis that you can separate specific drivers.**

See above response.

**Page 16 line 3-6: Connect this sentence to the following paragraph.**

Thanks. We will connect the two paragraphs in the revised manuscript.

**Page 16 line 17: Is the "framework" the non-linear component? Maybe just call it that ? non-linear, rather than a framework. This implies a more complex process.**

As we have explained in our response to Referee #1, it is not only the non-linear component that has been applied in this study. Our approach consists of several components including data collection, feature construction, non-linear machine learning algorithms and Granger causality analyses. We will try to make it more clear

in the revised manuscript.

**Page 17 line 11: explain "feature construction"**

With the term 'feature construction' we mean the process that is followed in order to extract informative predictors for a model. In our case, predictors have been extracted from the raw time series using domain knowledge and include climatic cumulative/lagged variables and climate extreme indices. This term is explained in detail in Sect. 3.3 of the manuscript.

**Page 17 line 16: update word order to read "causality based approaches indicate"**

Yes, thanks.

---

## Author Response (AR1)

**Authors' response to the Referees**

For clarifying our answers to the reviewers' comments, the following scheme is used: comments of the reviewers are denoted in **bold** font, our answers are denoted in plain font and quotes from the manuscript are denoted in *italic*. We also number the different comments of the reviewers as: I.J, where I is the number of the referee and J the number of the remark.

**Anonymous Referee #1**

**The manuscript introduces a Granger causal inference approach to investigate climate-vegetation dynamics. A great effort in collecting a representative enough dataset has been pursued to study such dependencies. The authors put emphasis on the non-linearity of the approach since the VAR method typically used in the canonical Granger approach is here replaced by a non-linear regression tool, the random forests method. Authors claim that the causal patterns are more clearly identifiable than with traditional linear models. Overall, I think this is a very nice piece of work that is worth publishing after some clarifications and addressing some problems.**

We would like to thank the reviewer for the appreciation of the manuscript, for the constructive feedback, and for the thorough assessment. Below we provide a point-to-point response to each comment and we present the changes in the revised version of the manuscript.

**Below authors will find a long list of minor and major comments that I hope they can address.**
**- 1.1) abstract: 3: unravel the influence... : this looks like an ambitious goal that I'm not sure authors finally managed to address**

We agree with the reviewer and changed the sentence by stating that the technique "allows to further unravel the influence". The sentence now reads *'Data of this kind [Earth observations] provide new means to further unravel the influence of climate on vegetation dynamics'*.

**- 1.2) 4: existing statistical methods: do authors refer to linear ones only, right?**

'Existing' has been replaced by 'commonly-used' in the revised manuscript.
    Abstract: 3: *However, as advocated in this article, commonly-used statistical methods that assume linearity are often too simplistic to represent complex climate-vegetation relationships.*

**- 1.3) 8: (also in the title) the word 'framework? looks too ambitious.**

**In the end, authors only proposed to follow the Granger approach with a different feature selection and regression method. Does this qualify to call it framework?**

We understand the concern, yet 'framework' can be defined as 'a basic conceptional structure (as of ideas)' (Merriam-Webster dictionary). We chose to use the word 'framework', because we believe it reflects well the conceptional structure followed here, which goes beyond adopting the Granger approach with a different feature selection and regression method, and consists of several sequential steps. First, multiple datasets of the most important climatic variables have been collected and converted into a common temporal and spatial resolution (a multidimensional data-cube). Second, by applying feature extraction techniques and domain knowledge, predictor variables have been constructed. Third, a non-linear machine learning algorithm has been designed and applied. Fourth, causality has been assessed based on Granger causality. As such, we think that we propose a complete framework that can be used for knowledge discovery in climate sciences, and which in this paper is applied to unveil climate-vegetation dynamics, but that could be used to detect other causal patterns in the climate system. Moreover, by using non-linear models and feature extraction techniques, we argue that this framework substantially differs from methodologies that are common practice in the field. For these reasons, we opted to keep the word 'framework' in the revised manuscript.

**1.4) p2.29: y alludes to the NDVI time series: shouldn't be the IAV of NDVI thereof?**

It is true that we finally model the IAV of NDVI, this is just a starting point of explaining the basic model. We have added the word 'anomalies' in order to clarify which is the target variable.

p2.31: *In this work $\boldsymbol{y}$ alludes to the NDVI anomalies time series at a given pixel, whereas $\boldsymbol{x}$ can represent the time series of any climatic variable at that pixel (e.g. temperature, precipitation or radiation).*

**1.5) p3.7: for me, describing the $R^2$ is too verbose and useless in a scientific journal nowadays**

The $R^2$ is indeed a well-known performance measure, we wrote down the formula just to avoid confusion: for linear models one often computes the $R^2$ as a correlation coefficient, while for non-linear models one has to compute the $R^2$ in a different way, using the formula in the manuscript. This is perhaps obvious, but it might be useful to have the formula in the paper for readers that are less familiar with the different definitions of $R^2$. We decided to keep this formula in the revised manuscript. We also stress that the $R^2$ is computed on data that was not used for model training, so there is no need for adjustments to account for the degrees of freedom of the models, like is commonly done in statistics.

**1.6) p3.eq2-3: the $\approx$ symbol is meaningless here. I'd suggest to include the signal model here ($y = \hat{y} + e$), and describe the assumptions about the noise model (Gaussian, uncorrelated?). Also, I don't find natural that both eqs. have the same model coefficients $\beta_{11p}$.**

We agree. We have incorporated these changes in the revised manuscript.
p3.21:

$$y_t \;=\; \hat{y}_t + \epsilon_1 = \beta_{01} + \sum_{p=1}^{P} \left( \beta_{11p} y_{t-p} + \beta_{12p} x_{t-p} \right) + \epsilon_1 \qquad (1)$$

$$y_t \;=\; \hat{y}_t + \epsilon_1 = \beta_{01} + \sum_{p=1}^{P} \beta_{11p} y_{t-p} + \epsilon_1 \qquad (2)$$

*with $\beta_{ij}$ being parameters that need to be estimated and $\epsilon_1$ and $\epsilon_2$ referring to two white noise error terms.*

**1.7) p3.27: authors should clarify the sentence "neither variables nor observational ... and errors are ...". Independent of what? each other? independent noise? Please be explicit and consistent in the use of the terms 'error', 'noise', 'residuals'.**

We meant independent from each other. We added the phrase "from each other" to the revised manuscript.
p3.27: *However, our above definition differs from the perspective in research papers that develop statistical tests for Granger causality (Hacker and Hatemi-J, 2006), because we intend to move away from statistical hypothesis testing, since the assumptions behind such testing are typically violated when working with climate data where neither variables nor observational techniques are fully independent from each other in most cases, and errors are not normally distributed (see Sect. 2.4 for a further discussion).*

**1.8) p4.10, eq4: describe the meaning of $\beta_{13}$ and all terms involved in the equation**

We have included a more extensive explanation of the tri-variate extension of Granger causality in the revised manuscript.
p4.15: *As previously mentioned, the time series $\boldsymbol{w}$ may also have a causal effect on $\boldsymbol{y}$ and be correlated with $\boldsymbol{x}$. For this reason, $\boldsymbol{w}$ should be included in both models (baseline and full), so that the method can cope with cross-correlations between climatic drivers of vegetation anomalies.*

**1.9) p4.26: Maybe I'm missing something but if you split the data this way, aren't you discarding long-term correlations. Also, by simple xval, results depend to a large extent of the selected data splits. To avoid this, why not LOO?**

Our motivation for doing 5-fold cross-validation instead of leave-one-out (LOO) was mainly motivated by computational reasons. LOO takes a long time to compute and is generally not the recommended method when analyzing large datasets (Elisseeff and Pontil, 2003). As we are working with an extremely large dataset here, computational efficiency is always the first criterion to look for. For this reason, we keep the same evaluation procedure in the revised manuscript.

Elisseeff, A., & Pontil, M. (2003). Leave-one-out error and stability of learning algorithms with applications. NATO science series sub series iii computer and systems sciences, 190, 111-130.

**1.10) p5.10: the same comment about the $\approx$ symbol before: please include the signal model equations here too.**

We have changed it in the revised manuscript.
p5.14:

$$y_t \quad = \quad \hat{y}_t + \epsilon_1 = \beta_{01} + \sum_{p=1}^{P} \left( \beta_{11p} y_{t-p} + \beta_{12p} x_{t-p} + \beta_{13p} w_{t-p} \right) + \epsilon_1 \quad (3)$$

$$x_t \quad = \quad \hat{x}_t + \epsilon_2 = \beta_{02} + \sum_{p=1}^{P} \left( \beta_{21p} y_{t-p} + \beta_{22p} x_{t-p} + \beta_{23p} w_{t-p} \right) + \epsilon_2 \quad (4)$$

$$w_t \quad = \quad \hat{w}_t + \epsilon_3 = \beta_{03} + \sum_{p=1}^{P} \left( \beta_{31p} y_{t-p} + \beta_{32p} x_{t-p} + \beta_{33p} w_{t-p} \right) + \epsilon_3 \quad (5)$$

**1.11) p5.15: formally it is straightforward, but not computationally or for decision making which may be an infeasible problem.**

Indeed. We only want to convey here that the formal definition of Granger causality does not change in the case of more than three time series.

**1.12) p6.1-3: if you want to keep this statement, please discuss about the theoretical implications, and cite other non-linear Granger causality methods (a simple search in Google will return you several dozens of works in machine learning, kernel methods, time series forecasting, econometrics and finance).**

We agree that there is previous work on non-linear Granger causality. These methods typically assume stationarity in the time series, and they are hence not immediately applicable for climatic time series. We have extended the paragraph with a more thorough discussion on related work and new references to these articles. We also clarify that we have not introduced non-linear Granger causality for the first time, yet, to our knowledge, more complex methods that use

Granger causality have not been widely applied in the field of climate sciences. We refer to the literature cited in Sect. 2.3.

p6.2: *In other fields, such as in neurosciences, kernel methods or other non-linear models have been used for the investigation of non-linear Granger causality relationships between time series (Marinazzo et al., 2008; Ancona et al., 2004). In our analysis, we stick to simple non-linear methods that are applicable to large datasets. More sophisticated approaches typically do not scale well enough in global climate-vegetation datasets.*

**1.13) p6.1-14: verbose, remove or summarize a lot.**

This might be obvious for a well-informed reader, but we believe that an explanation of that kind is needed for readers that are less familiar with Granger causality and time series forecasting. We therefore decided to keep this paragraph as it is.

**1.14) p9.eq: the upperscript T may confuse as in standard algebra that symbol stands for transpose.**

We have replaced the symbol $T$ with $T_r$ in order to make it clearer in the revised manuscript.
p10.20:

$$y_t \approx y_t^{T_r} = \alpha_0 + \alpha_1 \tag{6}$$

**1.15) p9.3: obvious non-stationary: sometimes it is not that obvious.**

We deleted the word 'obvious' in the revised manuscript.

**1.16) p12.6: a sentence does not conform a paragraph.**

We tend to disagree, a paragraph is 'a subdivision of a written composition that consists of one or more sentences' (again from the Merriam-Webster). We decided to keep it a stand-alone sentence in order to highlight it as a conclusion from the entire section

**1.17) And by the way... is 1 degree enough resolution to claim something about causation? do the expected relations occur at such broad scale?**

Most atmospheric variables change consistently at spatial resolutions that are even coarser than 1 degree; in fact most current climate models resolve the land-atmospheric interactions while working at coarser resolutions. We also note that there is a trade-off between spatial resolution and time period covered by the datasets. The 1-degree resolution is a characteristic of the datasets we are working with, and if we wanted to focus on finer resolutions, we would need to incorporate datasets from sensors covering more recent years only, thus multi-decadal analysis would not be possible. The 1-degree resolution, in addition, still also allows us to perform our calculations in a reasonable amount of time.

**1.18) p12.13: please avoid overoptimistic phrases like "our non-linear random forestS".**

It has been rephrased to "the non-linear random forest model" in the revised manuscript.
p14.9: *To analyze the effect of climate on vegetation more thoroughly, we substitute the linear ridge regression model (VAR) by the non-linear random forest model.*

**1.19) p12.17: "simple correlations" should be "spurious correlations"? in any case this sentences deserves more clarification and be more explicit Fig4: some discussions and words of caution should be given about deriving conclusions out of $R^2 \sim 0.4$. By the way, why the maximum in the scale is not explicit for $R^2$ and you select that threshold in 0.4? Why not using the statistical significance of the correlation rather than the $R^2$ score? Can authors include and discuss the maps of R p-values?**

As mentioned in the manuscript, the assumptions of common statistical tests are violated due to the non-stationarity of the data and the non-linearity of the proposed model. Developing a statistical test that is able to handle non-stationary time series and non-linear models is not a trivial task. As far as we know, no such test exists. Therefore, we decided to focus on expressing Granger causality in a quantitative way instead of a qualitative way, and stress the gained improvement with the use of a non-linear model. We have included the relevant references and a more thorough discussion about existing statistical tests in Section 2.4 of the revised manuscript.
p7.1: *See entire Section 2.4.*

**1.20) Fig4 caption: 'with respect to a the' to be corrected**

Done.
Fig4 caption: *Improvement in terms of $R^2$ by the full ridge regression model with respect to the baseline ridge regression model that uses only past values of NDVI anomalies as predictors;*

**1.21) p13.3: 'our'?**

We replaced "our" with "the" in the revised manuscript.

**1.22) p14.3: what are these patterns of the explained variance? some clarification is needed here? I guess authors refer to spatial patterns of variation? If that is the case, it looks not really obvious to talk about spatial relations when no such relations are considered to build up the regression models.**

In fact this section does not refer to spatial patterns, but to a general improvement of the full model versus the restricted model. We removed the word "patterns" from the referred statement in the revised manuscript.

**1.23) p14.7: unambiguous? some more comments are needed, and if possible supported by numerical scores.**

With unambiguous we just mean that the improvement is clearly visible here (in the order of 20 to 60%). This claim is supported by Fig. 5b. So we opted to keep this word.

**1.24) p13-14: as a reader I'd prefer to have in the same figure panel the current figures 4 and 5 so I could directly compare results in one shot.**

We understand this concern and understand that a 3 × 2 figure would appear somehow more convenient. However, we chose to have Fig. 5 in a separate panel because this is the main figure of the paper.

**1.25) p15.3: what do authors mean by 'higher-lever variables'? are you thinking of higher-order statistical relations between variables? this is absolutely confusing.**

We agree. With the term 'higher-level variables' we refer to the past cumulative climate, lagged variables and climate extreme indices that are considered as predictor variables. This has now been clarified in Section 3.3 of the revised manuscript.

p12.1: *We do not limit our approach to considering raw versus anomaly time series of the data sets in Table 1 as predictors, but also take into consideration different lag times, past-time cumulative values and extreme indices. These additional predictors, further referred to as higher-level variables, are calculated based on raw and anomaly time series.*

**1.26) p15.5: please provide a copy of the (Papagiannopolou et al, in review) so reviewers can appreciate differences in approaches and results. Alternative, cite an accessible work to support the claims in this paper.**

The referred article is enclosed. This should be made available to the editorial and reviewers only.

**1.27) p16.3-18: please clarify these paragraphs in several ways: 1) the spatial encoding is not at all clear since typically the input (feature) space is augmented with the neighbors which are then used to predict on the central pixel (the length of the observation variable does not change), which seems not to be the case here. 2) it is weird that the spatial info didn't improve the results: I'd thank the authors to include such 'negative results' but then some comments and clarifications are needed (e.g. 1 degree is already integrating too much info, or spatial encoding was not taking into account pixel spatiotemporal variances?)**

Yes, the approach we followed is as described by the reviewer. The feature space of one pixel is augmented with the features of the 8 neighboring pixels. We also expected to see a more substantial improvement using spatial information but this is not the case. We have extended the discussion in page 18 of the revised manuscript to hypothesize a reason for this limited improvement.

p18.20: *A possible explanation for this result is that the model without the spatial information cannot be outperformed because of the large dimensionality of the feature space, which may include redundant information, in combination with the low number of observations per pixel (Fig. 5a). Note that in this case the number of observations per pixel remains the same as in the original model (360 observations) while the number of predictor variables is 9 times larger.*

**1.28) p17.9: as said before I feel claiming a 'novel framework? is far too much for this contribution.**

See response 1.3.

**1.29) p17.15-20: some claims are contained here without empirical justification. I think that authors lost a nice opportunity here to explain the causal relations. For example, to me it seems ad hoc to justify results with a simple 'the predictive power of the model is especially high in water-limited regions'. Probably this is true but some numbers are needed to support it. I suggest to include a summarizing feature ranking of the LR vs RFs (e.g. permutation analysis, and surrogate analysis). Also, summarize results per regions and biomes would help discussing the results more profoundly, elevating the debate. Of course, these two issues may require some more work, but I sincerely think they are mandatory to make a sound publication.**

Actually, we have performed this kind of analyses, taking feature rankings using RFs. However, these rankings become unstable due to highly-correlated predictors. A specialized approach would be needed here, in which groups of features are ranked instead of individual features. This makes the rankings more stable

and improves the interpretability. It is exactly what we do in the complementary paper (Papagiannopoulou et al., under review). We also agree with the reviewer that a stratification of the results according to regions/biomes is a relevant addition to the paper. The revised version provides the results stratified according to IGBP land cover classes for both the baseline and the full random forest model. These new results are discussed in Section 4.2 and a new figure, Fig. 6 has been added.

p15.20: *For a better understanding of the results obtained by the two models, we average the performance of each model regionally. More specifically, we use the International Geosphere-Biosphere Program (IGBP) (Loveland and Belward, 1997) land cover classification to stratify the mean and variance of $R^2$ for both the baseline and the full model in Fig. 5 per IGBP land cover class. The barplot in Fig. 6 shows that the full model outperforms the baseline model in all IGBP land cover classes, i.e. that Granger causality exists for all these biomes. In the parentheses we note the number of pixels per region. The error bars indicate that the variances of the two models are analogous, i.e. they are low or high in both models in the same land cover class. For the Closed Shrublands region, one can observe the highest difference between the two models, yet only 19 pixels belong to this biome type. In savanna regions, the performance of the full model is high in comparison with other regions (see Fig. 5). On the other hand, the lowest performance improvement of the full model with respect to the baseline is observed for the regions of Deciduous Needleleaf Forests and Evergreen Broadleaf Forests. This shows that for these two regions climate is not identified as a major control over vegetation dynamics (see discussion in previous paragraph about tropical and boreal regions).*

**1.30) p18.8: reproducibility is not possible as data is not available yet. do authors plan to make these data available to the community?**

All codes are freely available and documented on GitHub and will comply with the Copernicus data policy. On the other hand, the full database is formed by a collection of datasets that are all publicly available and that, due to copyright conflicts, cannot be openly distributed. The relevant link is provided in Section 6.

**Anonymous Referee #2**

**General Comments:**
**Reviewer summary: The manuscript presents a non-linear Granger causality analysis to investigate climate-vegetation interactions. Anomalies of the normalized vegetation index (NDVI) are analyzed in conjunction with a full set of climate variables taken from re-analysis, in situ, and satellite observations. The data provide multi-decadal global coverage for water availability (precipitation, snow water equivalent and soil moisture data), temperature, and radiation. All data spans the period 1981-2010 at the global scale and has been converted to**

a common monthly temporal resolution and 1×1 degree spatial resolution. At each pixel the NDVI data is considered the response and the climate data the predictor variables. A moving window of twelve months is used to determine if the climate data Granger-causes the NDVI value. Analysis is performed on NDVI anomalies computed by subtracting the corresponding monthly expectation from the detrended time series. The climate data as well as cumulative values and extreme indices calculated from the climate data were included as predictor variables. The non-linear Granger causality uses a non-linear random forest model, and is shown to explain more of the variance than the linear Granger analysis.

Article contribution and overall impact: This study makes an effort to use multiple climate data sources to tease out predictability for vegetation anomalies. The authors highlight improvements with the non-linear method compared to traditional Granger causality, as well as the importance of using extreme events. The discussion would benefit from a more explicit discussion of the uncertainty associated with the climate datasets used as predictors. Given that this study precedes or supports Papagiannopoulou et al (in review), more discussion of those results and their importance would be useful as that study is not available to the reader. Specifically, the follow-on study highlights the importance of specific climate predictors for particular regions. It is not clear how those variables are chosen from the many climate predictors, and it would be useful to provide an example in this manuscript to highlight the strength of this method with a clear detailed regional example.

We would like to thank the reviewer for the feedback, and the thorough assessment of the manuscript.

We agree that the study Papagiannopoulou et al. (in review), in which we apply the method to discern the importance of different climatic drivers, may be useful for the referees to assess the potential of our framework. As we mentioned in our response to the Anonymous Referee #1, that article is enclosed in the resubmission of the revised paper, so it can be available to the editor and reviewers. As the referred article is a follow-up from this GMD paper, as the referee states, we do not see the need to provide details about its specific results within the GMD paper.

Below we provide our pointwise response to the review, as well as the changes in the revised manuscript.

**Detailed comments:**
**2.1) Page 1 line 17-18: Should this read "predictions of vegetation in response to future climate can be improved through a better understanding..." ? as you are looking for climate drivers of vegetation.**

We think that the initial sentence *"Because of the strong two-way relation-*

*ship between terrestrial vegetation and climate variability, predictions of future climate can be improved through a better understanding..."* is in fact correct. In this paragraph, we discuss the complex two-way interactive relationship between vegetation and climate in order to state the importance of understanding climate dynamics to predict climate accurately. Therefore, a better understanding of the vegetation response to past climate variability, brings us one step further in understanding future climate, since the latter will also be affected by the fate of vegetation. Therefore, we opted to keep the sentence as it is.

**2.2) Page 2 line 22: define "higher-level features" here and throughout manuscript. It is not clear what these are. (Pg 11 line 4, pg.15 line 2)**

See response 1.25.

**2.3) Page 2 line 24: define "higher-level climate variables" not clear what this is.**

See response 1.25.

**2.4) Page 3 line 2-7: May not be necessary to include full definition of $R^2$.**

See response 1.5.

**2.5) Page 3 line 30: update "might lead to wrong" to "might lead to incorrect"**

We have changed this phrase in the revised manuscript.
p4.4: *However, such an analysis might lead to incorrect conclusions, because additional (confounding) effects exerted by other climatic or environmental variables are not taken into account (Geiger et al, 2015).*

**2.6) Page 12 line 15-23: Are the results for all variables, or the most predictive variable, or a set of variables at each pixel?**

We use all the variables at each pixel in order to obtain the results presented in this section.
This is stated in Section 4.1.

**2.7) Page 12 line 26-27: Why is this chosen as the minimum? Please explain or provide citation.**

This statement comes from the definition of Granger causality. The minimum explained variance can be achieved by using the history of the target variable only, and this is basically the model referred to as 'baseline model' in

the manuscript.

**2.8) Page 13 line 10: by what margin is the uncertainty larger in these regions, and for what reasons?**

As one can notice from the map in Figure 5b, the improvement of the full model in terms of $R^2$ compared to the baseline is low in these regions. Therefore, the results indicate that the Granger causal effects of climate on vegetation anomalies in these areas are not obvious. This is why we enumerate a set of studies which explore the main drivers of vegetation in these regions, explaining the poor predictive performance of the full model with respect to the baseline model.

**- 2.9) Are you referring to all the climate variables, if not please qualify.**

Yes, we are referring to all the climate variables included in the dataset.

**- 2.10) The citation references error for soil moisture.**

Corrected.
*Dorigo, W., Wagner, W., and et al.: ESA CCI Soil Moisture for improved Earth System understanding: state-of-the-art and future directions, Remote Sensing of Environment, in review.*

**- 2.11) Add citations, which support the amount of uncertainty in these regions for the remaining data types.**

See response 2.8.

**2.12) Page 13 line 7 to bottom and page 14 line 1-4: Move this to discussion.**

The subject of this section is to present the results obtained from the proposed methodology. The part the reviewer refers to discusses the performance of the proposed methodology in comparison to other models. However, we understand the fact that the presentation of additional results in the "Discussion" section is a bit confusing. Therefore we decided to combine the two sections in one with the name "Results and discussion".

**2.13) Page 14 line 1: Update to "vegetation anomalies are not necessarily"**

We have added the word 'anomalies' here in the revised manuscript.
p15.7: *(b) these are regions in which vegetation anomalies are not necessarily primarily controlled by climate, but may also be driven by phenological and bi-*

*otic factors*

**2.14) Page 14 line 7: Use different phrasing for "unambiguous"**

By "unambiguous" we just mean that the improvement is clearly visible here, which we believe reflects correctly the meaning of this word. We kept the word in the revised manuscript.

**2.15) Page 14 line 7-12: move to discussion.**

See response 2.12.

**2.16) Page 14 line 8-10: Recommend re-wording this. The limit for figure 5 and the presentation of the non-linear analysis is still to a limit of $R^2 = 0.4$ as in figure 4? An $R^2$ of 0.4 does not seem like a strong correlation. Though figure 5 is improved from figure 4 there are large portions that show no improvement, and the overall explained variance is below 40% in most regions.**

We kept the same colorbar scale in both figures for a better figure-to-figure comparison. We note that $R^2$ is not to be mistaken by 'correlation' in the non-linear models; it is a performance measure that indicates how close the model predictions are to the real values of the target variable. In our study, we remove the seasonal cycle from the NDVI time series and we target the NDVI anomalies, making the task more difficult than predicting the raw NDVI time series, since the autocorrelation in the NDVI anomalies time series is much lower. Note that if we target the raw NDVI time series (which includes the seasonal component), the $R^2$ is close to 1 in most of the regions (see Fig. 7 in the manuscript). In addition, it is worth noting that there are other factors such as fires or harvest that affect vegetation dynamics but are not included in the dataset, as mentioned in the discussions. Therefore, we should be aware that we focus on explaining the variance of the NDVI anomalies, taking into account only climatic variables, and focusing on the part of their explanatory power that truly reflects Granger-causal relationships.

**2.17) Page 14 line 10: "comparison between figs 4b and 5b" explain in more detail. It would be easier for the reader to compare these if they were in one figure block, or on the same page.**

See response 1.24.

**2.18) Page 15 line 5: Please provide more detail about this study. It comes up frequently in the manuscript, and a larger summary with details (supportive numbers or examples from regions) would be helpful since we do not have access to the manuscript.**

See response 1.26.

**2.19) Page 15 line 11: Has a test been run with only the anomalies and extremes? Would that sub-set of predictors provide strong predictive performance?**

We agree with the reviewer. We can include more experimental results at this point in order to figure out the importance of anomalies and extremes in particular. In the paper Papagiannopoulou et al. (in review), there is a thorough discussion about the different groups of variables that have been tested for Granger causality. Results of isolating the impact of extremes are also presented in that same manuscript. Again, this paper is enclosed in the resubmission.

**2.20) Page 16 line 1-2: Provide more detail from supporting manuscript for current manuscript. It is necessary to support this analysis that you can separate specific drivers.**

See response 1.26.

**2.21) Page 16 line 3-6: Connect this sentence to the following paragraph.**

We have connected the two paragraphs in the revised manuscript by adding the following sentence:
p18.8: *In addition, it is quite likely that neighboring areas have similar climatic conditions, which in their turn affect vegetation dynamics in a similar manner.*

**2.22) Page 16 line 17: Is the "framework" the non-linear component? Maybe just call it that ? non-linear, rather than a framework. This implies a more complex process.**

See response 1.3.

**2.23) Page 17 line 11: explain "feature construction"**

With the term 'feature construction' we refer to the process of extracting informative predictors for a model. In our case, predictors have been extracted from the raw time series using domain knowledge and include climatic cumulative/lagged variables and climate extreme indices. The explanation is given in the corresponding section (Section 3.3).

**2.24) Page 17 line 16: update word order to read "causality based approaches indicate"**

Corrected.

[revised manuscript text omitted]